# Fringe proteins modulate Notch-ligand *cis* and *trans* interactions to specify signaling states

Lauren LeBon[1,2], Tom V Lee[3], David Sprinzak[4], Hamed Jafar-Nejad[3], Michael B Elowitz[1,2]*

[1]Howard Hughes Medical Institute, California Institute of Technology, Pasadena, United States; [2]Division of Biology and Biological Engineering, California Institute of Technology, Pasadena, United States; [3]Department of Molecular and Human Genetics, Baylor College of Medicine, Houston, United States; [4]Department of Biochemistry and Molecular Biology, Tel Aviv University, Tel Aviv, Israel

**Abstract** The Notch signaling pathway consists of multiple types of receptors and ligands, whose interactions can be tuned by Fringe glycosyltransferases. A major challenge is to determine how these components control the specificity and directionality of Notch signaling in developmental contexts. Here, we analyzed same-cell (*cis*) Notch-ligand interactions for Notch1, Dll1, and Jag1, and their dependence on Fringe protein expression in mammalian cells. We found that Dll1 and Jag1 can *cis*-inhibit Notch1, and Fringe proteins modulate these interactions in a way that parallels their effects on *trans* interactions. Fringe similarly modulated Notch-ligand *cis* interactions during *Drosophila* development. Based on these and previously identified interactions, we show how the design of the Notch signaling pathway leads to a restricted repertoire of signaling states that promote heterotypic signaling between distinct cell types, providing insight into the design principles of the Notch signaling system, and the specific developmental process of *Drosophila* dorsal-ventral boundary formation.

*For correspondence: melowitz@caltech.edu

**Competing interests:** The authors declare that no competing interests exist.

**Reviewing editor**: Helen McNeill, The Samuel Lunenfeld Research Institute, Canada

## Introduction

The Notch signaling pathway mediates communication between adjacent cells (*Artavanis-Tsakonas et al., 1999*). As the primary juxtacrine signaling pathway, Notch carries out the fine-detail work of animal development, from drawing sharp boundaries between cell populations, to laying out checker-board-like lateral inhibition patterns across a tissue, to controlling fractal-like branching structures in vascular and lymphatic systems (*Lewis, 1998*; *Artavanis-Tsakonas et al., 1999*; *Irvine, 1999*; *Bray, 2006*; *Phng and Gerhardt, 2009*).

Notch signaling occurs when a DSL (Delta/Serrate/Lag2) ligand binds to a Notch receptor on a neighboring cell, triggering proteolytic cleavage of the Notch receptor and endocytosis of the Notch extracellular domain into the signal-sending cell (*Fortini, 2009*). This mechanism releases the Notch intracellular domain (NICD), which translocates to the nucleus and interacts with the CSL complex (CBF1/Suppressor of Hairless/Lag1; also known as RBP-Jκ) to initiate transcription of target genes (*Artavanis-Tsakonas et al., 1999*; *Fortini, 2009*). In addition, Notch receptors and DSL ligands interact within the same cell in a process called *cis*-inhibition (*del Alamo et al., 2011*). Overexpression and loss of function studies have revealed that DSL ligands can *cis*-inhibit the ability of a cell to receive a Notch signal, and that this effect depends on the interaction of the extracellular domains of the receptors and ligands in the same cell (*Jacobsen et al., 1998*; *D'Souza et al., 2008*). Additionally, Notch receptors can reciprocally block DSL ligands in the same cell from sending signal (*Becam et al., 2010*;

**eLife digest** In animals, cells use a process called Notch signaling to communicate with neighboring cells. During this process, a protein known as a DSL ligand from one cell binds to a protein called a Notch receptor on a neighboring cell. This triggers a series of events in the neighboring cell that change how the genes in this cell are expressed. Notch signaling is involved in many processes including the early growth of embryos, the formation of organs and limbs, and the maintenance of stem cells throughout adult life.

Enzymes called Fringe enzymes can control Notch signaling by blocking or promoting the formation of the DSL ligand-Notch receptor pairs. It is also possible for a DSL ligand and a Notch receptor from the same cell to interact. This is thought to be important because it prevents an individual cell from sending and receiving Notch signals at the same time.

There are several different DSL ligands, Notch receptors and Fringe enzymes, so it is difficult to determine which configurations of receptors, ligands and Fringe enzymes can enable Notch signals to be sent or received. To address this problem, LeBon et al. investigated how Fringe enzymes acted on several different DSL-Notch receptor pairs in mammalian cells, and also in fruit flies. They focused in particular on the interactions that occurred within the same cell, as the role of Fringe enzymes in this type of interaction has not been examined previously.

The experiments revealed that Fringe proteins modify specific same-cell interactions in a way that enables a cell to receive one type of Notch signal from a neighboring cell and send a different type of Notch signal to another cell at the same time. More generally, these results show how an unconventional, 'bottom-up' approach can reveal the design principles of cell signaling systems, and suggest that it should now be possible to use these principles to try to understand which cell types send signals to which other cell types in many different contexts.

*del Alamo and Schweisguth, 2009*). Previous in vitro studies support a simple model in which ligand *cis*-inhibition of receptors and receptor *cis*-inhibition of ligands represent a single process, with ligands and receptors in the same cell forming an inactive complex that prevents them both from interacting in *trans* with neighboring cells (*Sprinzak et al., 2010*).

Recent work has suggested that these mutually inhibitory *cis* interactions between receptors and ligands in Notch and other signaling pathways can play a critical role in cell signaling (*Yaron and Sprinzak, 2012*). To illustrate, we analyze the Notch signaling state of a cell, defined by the cell's quantitative ability of a cell to send or receive signal using a given ligand. We consider a cell expressing one type of ligand and one type of Notch receptor. If the cell produces more receptor than ligand, *cis* interactions efficiently remove most or all ligand but leave an excess of free receptor, enabling the cell to receive, but not send, Notch signals (*Figure 1A*, top left). On the other hand, if the cell produces more ligand than receptor, *cis* interactions sequester the receptor, leaving an excess of free ligand, and permitting the cell to send, but not receive, signals (*Figure 1A*, top right). In this simple case, the relative levels of ligand and receptor expression produce a sharp threshold between sending and receiving signaling states and thereby regulate the strength and direction of signaling between neighboring cells (*Sprinzak et al., 2010*, *2011*). Consistent with the ratiometric nature of this model, many Notch-dependent developmental processes are highly sensitive to changes in receptor and ligand gene dosage, and show haploinsufficient mutant phenotypes (*de Celis et al., 1996*; *de Celis and Bray, 2000*; *Duarte et al., 2004*; *Phng and Gerhardt, 2009*; *Sprinzak et al., 2011*).

With only a single type of ligand and a single type of receptor it is relatively straightforward to evaluate signaling states (*Figure 1A*). However, in *Drosophila*, there are two DSL ligands, Delta and Serrate, while in mammals, there are four Notch receptors (Notch 1–4) and five canonical Notch ligands, three members of the Delta family (Dll1, Dll3, Dll4) and two members of the Serrate family (Jag1 and Jag2, homologues of *Drosophila* Serrate) (*Bray, 2006*; *D'Souza et al., 2008*). Each ligand–receptor pair can have a different interaction strength. For example, Dll4 interacts more strongly with Notch1 in *trans* than Dll1 (*Andrawes et al., 2013*). Moreover, in vertebrates, Notch signaling processes typically utilize combinations of multiple receptors and ligands. For example, during angiogenesis, the sprouting of new blood vessels depends on complex spatial expression of Notch1, Dll4, and Jag1 (*Benedito et al., 2009*; *Phng and Gerhardt, 2009*). In chick spinal cord development, generation of the

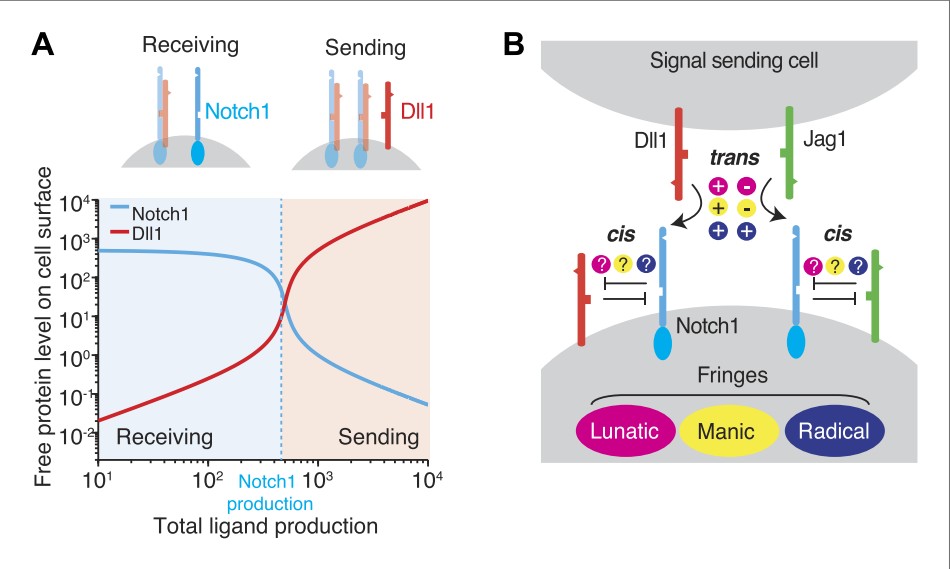

**Figure 1**. *Cis* interactions between receptors and ligands lead to exclusive sending and receiving signaling states. (**A**) In the blue shaded region, receptor expression exceeds ligand expression (as indicated schematically above plot), so that mutual *cis* interactions leave mainly free receptors, allowing the cell to receive, but not efficiently send, signals. When ligand expression exceeds Notch expression, mutual *cis* interactions consume most of the Notch receptors, leaving an excess of free ligand, favoring sending over receiving. (**B**) There are multiple potential ways in which Notch1 could interact in *cis* and *trans* with Jag1 and Dll1 ligands, and in which Fringe proteins could modulate these interactions. Known interactions are indicated by + and − for positive and negative regulation, respectively. Unknown ways in which Fringe proteins could modulate these interactions are indicated by question marks.

six subtypes of sensory and motor neurons depends on distinct expression domains of Dll1 and Jag1 (*Marklund et al., 2010*). In these and other examples, co-expression of multiple ligands and receptors enables a large number of possible *cis* and *trans* interactions, making it difficult to determine which cells are communicating to which other cells through which receptors and ligands.

Further adding to the complexity, Fringe glycosyltransferases modulate the interaction between receptors and ligands (*Panin et al., 1997*; *Moloney et al., 2000*). Fringe proteins act in the Golgi to transfer *N*-acetylglucosamine (GlcNAc) to *O*-fucose-modified EGF repeats in the extracellular domain of Notch (*Bruckner et al., 2000*; *Moloney et al., 2000*). In *Drosophila* there is a single Fringe, while in mammals there are three homologues: Lunatic Fringe (Lfng), Manic Fringe (Mfng) and Radical Fringe (Rfng). In vitro co-culture experiments have revealed the differential effects of each Fringe on *trans*-activation from different DSL ligands (*Hicks et al., 2000*; *Ladi et al., 2005*; *Hou et al., 2012*). When Lfng or Mfng is ectopically expressed in a receiver cell, *trans* signaling from Dll1 ligands is enhanced, while *trans* signaling from Jag1 ligands is decreased. The effects of Lfng and Mfng in vertebrate systems resemble the effects of Fringe in *Drosophila*, which strengthens Delta signaling and inhibits Serrate signaling (*Panin et al., 1997*). On the other hand, Rfng increases the *trans* response to both ligands (*Ladi et al., 2005*). Despite this work, the effects of Fringe on *cis* interactions, if any, remain unknown. Given the central role of *cis* interactions in determining signaling states, it is therefore essential to determine whether and how Fringes influence these interactions.

In general, to determine the cell's signaling state requires knowledge of (1) the levels of ligands, receptors and Fringe proteins; (2) the interaction strengths in *cis* and *trans* for each ligand–receptor pair, and (3) how the Fringe proteins act individually and in concert to modulate *cis* and *trans* interactions. Data for (1) are increasingly available in different systems, but (2) and (3) have not been measured comprehensively. Such measurements could enable prediction of the directionality and cell type specificity of signaling from expression measurements in diverse processes.

To begin to obtain these measurements, we analyzed Notch-ligand *cis* interactions and their dependence on Fringe proteins in cell culture. We studied the Dll1-Notch1 and Jag1-Notch1 ligand–receptor

pairs, as these two ligands are frequently used simultaneously for signaling in the same tissue, and because clear differences in the effects of Fringe on these ligand–receptor pairs in *trans* have been established (*Figure 1B*) (*Hicks et al., 2000*; *Ladi et al., 2005*). To confirm that our measurements were qualitatively relevant for in vivo Notch-dependent processes, we tested our findings in a series of *Drosophila* mutants. Together, these data support a role for Fringe in modulating *cis* interactions in mammalian cells and *Drosophila*. These data constrain the set of possible signaling states for cells expressing multiple Notch pathway components, and support the idea that Notch pathway architecture fundamentally favors heterotypic signaling between cells in distinct signaling states.

## Results

### Availability assay enables measurement of *cis* interactions in single cells

We developed a cell culture-based assay system to measure the *cis* interactions between different ligand–receptor pairs (*Figure 2*). As a base cell line, we used CHO-K1 cells that support Notch signaling, but do not endogenously express Notch receptors and ligands. We constructed a stable CHO-K1 cell line constitutively expressing a 'diverted' Notch1 receptor, hN1(ΔICD)-Gal4$^{esn}$ (*Struhl and Adachi, 1998*; *Sprinzak et al., 2010*) where the intracellular domain of Notch1 is replaced with yeast Gal4. This receptor activates a UAS-reporter gene but not endogenous Notch targets. Next, we stably integrated tetracycline-inducible Dll1 or Jag1 ligands fused to the Cerulean fluorescent protein, allowing us to control and read out ligand expression with the inducer 4-epitetracycline (4-epiTc) and readout the expression level by cerulean fluorescence. We denote these cell lines 'Notch1+Dll1' and 'Notch1+Jag1', respectively (*Figure 2A,B*).

In order to analyze *cis* interactions in these cell lines, we induced ligand expression and measured the levels of free receptors and ligands on the cell surface. To detect Notch1 receptors, we incubated cells with saturating concentrations of a fragment of the Dll1 ligand fused to the Fc epitope (Dll1$^{ext}$-Fc, 'Materials and methods' and *Figure 3—figure supplement 1*). We established that this reagent specifically labels Notch receptors that are available to participate in *trans* interactions with DSL ligands, but does not label Notch-ligand *cis* interaction complexes (*Figure 2C,D*). Similarly, we used saturating concentrations of a Notch1 receptor fragment-Fc fusion protein (N1$^{ext}$-Fc) to specifically bind available ligands on the cell surface ('Materials and methods' and *Figure 3—figure supplement 1*). In both cases, we detected bound chimeric Fc ligands or receptors with a fluorescently labeled anti-Fc antibody. We used the parental Notch1 cell line to quantify receptor availability in the absence of *cis*-ligand, and cell lines expressing only Dll1-Cerulean or Jag1-Cerulean to quantify ligand availability in the absence of Notch1 (*Figure 3—figure supplement 1*). As a negative staining control, we incubated CHO-K1 cells with both the ligand and receptor availability assay reagents to establish background staining levels. We plated cells at low density to minimize *trans* interactions, and included only single, isolated cells in our analysis (*Figure 3—figure supplement 3*).

Using this assay, we first verified that receptor and ligand availability correlated with receiving and sending ability, respectively. For example, siRNA knockdown of the Notch receptor decreased Notch availability (*Figure 3—figure supplement 2A*) and reduced Notch activation by Dll1 ligands adhered to cell culture plate surfaces (*Figure 3—figure supplement 2B*). Likewise, inducing ligand expression with 4-epiTc led to both increased ligand availability (*Figure 3—figure supplement 2C*), and increased ability to *trans*-activate a Notch reporter cell line in a co-culture assay (*Figure 3—figure supplement 2D*).

In principle, the C-terminal fusion of CFP to the ligand could affect its trafficking or other aspects of its regulation, such as binding to proteins that interact with its PDZ domain (*Pintar et al., 2007*). We therefore compared the ability of untagged DSL ligands to *cis* inhibit Notch receptors with that of the CFP-tagged ligands. We transiently transfected either tagged or untagged ligands into our Notch receiver cells and measured the resulting decrease in receptor availability. We observed qualitatively similar results with untagged ligands as we did with the CFP fusions (*Figure 3—figure supplement 4*). Thus, although CFP fusions could affect other properties of DSL ligands, it does not appear to qualitatively affect the results shown here.

### Quantitative analysis of *cis*-inhibition by Dll1 and Jag1

Next, we compared the relative abilities of Dll1 and Jag1 to *cis*-inhibit Notch1. We induced varying levels of ligand expression in the Notch1+Dll1 and Notch1+Jag1 cell lines, and analyzed the cells with the availability assay. These experiments produced three key observations (*Figure 3*):

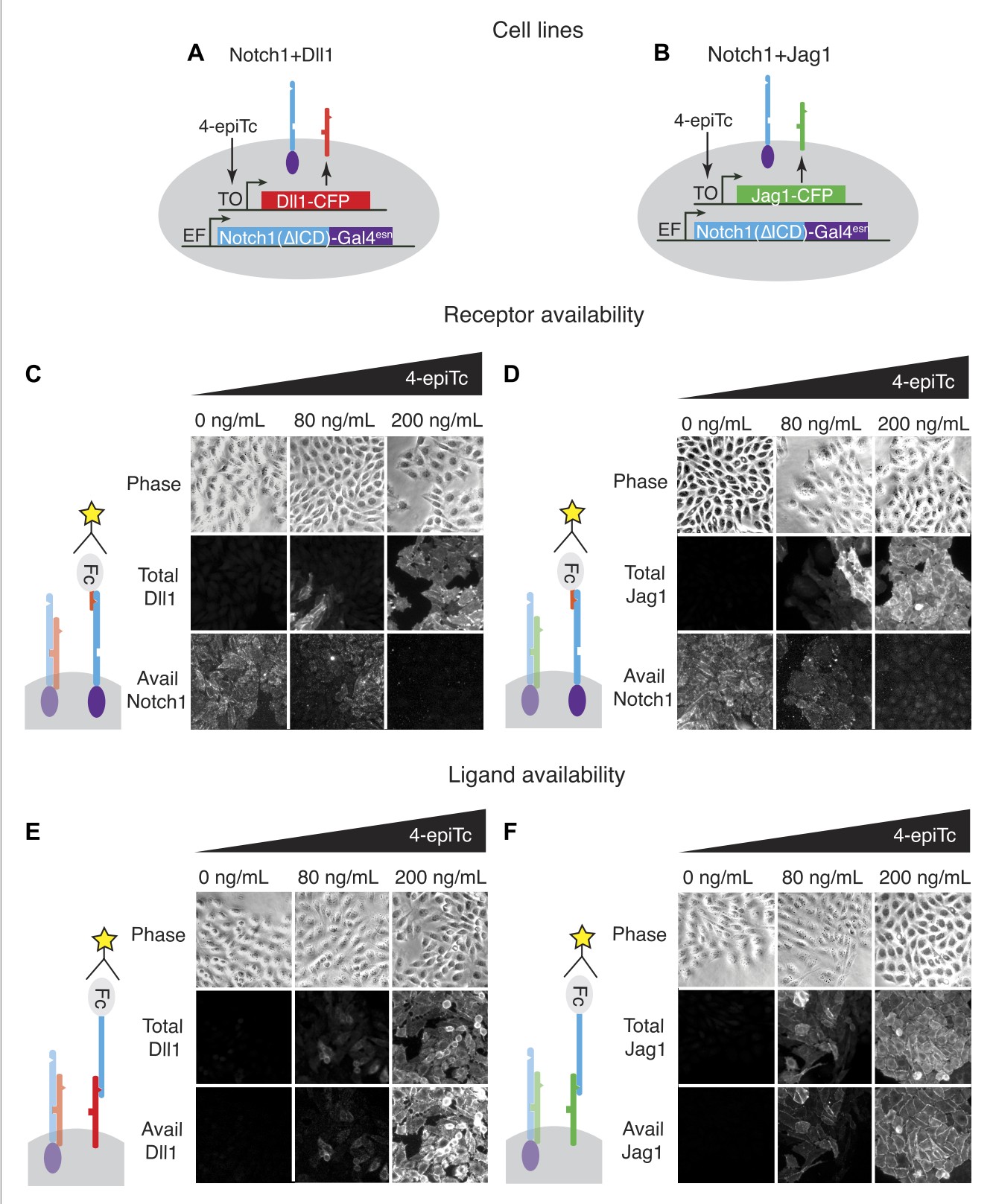

**Figure 2**. The availability assay labels receptors and ligands that can participate in *trans* signaling. (**A** and **B**) Stable CHO-K1 cell lines constitutively express a Notch1-Gal4 chimeric receptor and a tetracycline-inducible Dll1 (**A**) or Jag1 (**B**) ligand fused to cerulean fluorescent protein. (**C** and **D**) In the receptor availability assay, soluble Dll1ext-Fc is bound to free Notch receptor on the surface of live cells. After fixation, bound Dll1ext-Fc is labeled with

*Figure 2. Continued on next page*

*Figure 2. Continued*

anti-Fc fluorescent reagents. Increasing ligand-Cerulean expression reduces receptor availability, as shown in these snapshots (**C**, **D**, bottom panels). (E and F) The ligand availability assay works similarly, except soluble Notch1$^{ext}$-Fc fragments bind free ligands on the cell surface. Increasing ligand-Cerulean expression (**E**, **F**, middle panels), leads to increased ligand availability (**E**, **F** bottom panels). The surface ligand availability shows high spatial correlation with the total cellular ligand staining. Note that cells were plated at high cell density for illustration purposes. For quantitative analysis, cells were dissociated and plated at low density before staining (***Figure 3—figure supplement 3***).

First, both Dll1 and Jag1 can fully *cis*-inhibit Notch1 receptors (***Figure 2C,D***, ***Figure 3A,B***). Available Notch1 staining decreased to background levels in a dose-dependent fashion with increasing expression of either Dll1 or Jag1, indicating that both Dll1 and Jag1 ligands can fully reduce Notch availability. To confirm that this reduction in Notch availability was not an artifact of ligand overexpression, we compared wild-type Dll1 to Dll1-F199A, which contains a point mutation in the DSL domain that was previously shown to be partially deficient in *trans*-activation and *cis*-inhibition (***Cordle et al., 2008***). While transiently expressed wild-type Dll1-mCherry reduced Notch availability to background levels, the Dll1-F199A ligand showed only a partial reduction in Notch availability (***Figure 3C***).

Second, Dll1 appeared to be more potent than Jag1 as a *cis*-inhibitor of Notch1. We fit a simple model describing *cis* interactions (see ***Figure 1*** and 'Materials and methods') to the single cell data from each condition, and found that approximately twice as much available Jag1 on the cell surface was needed to reduce Notch availability by 50% compared to Dll1 (***Figure 3D***). Dll1 is also more potent than Jag1 as an activator of Notch1 in *trans* (***Figure 3—figure supplement 2D***), suggesting the possibility that *cis* and *trans* interaction strengths between a receptor and ligand could be correlated.

Third, Notch1 reduced both Dll1 and Jag1 availability, supporting the mutual inhibition model of *cis* interactions for both ligands (***Sprinzak et al., 2010***). In the Notch1+Jag1 and Notch1+Dll1 cell lines, we observed significantly reduced ligand availability compared to corresponding ligand-only cell lines at the same levels of total ligand expression. Moreover, this effect on available ligand could be rescued by knocking down expression of Notch1 with siRNA (***Figure 3E,F***). Because ligand availability correlates with sending ability (***Figure 3—figure supplement 2***), these results support a role for Notch1 in decreasing Dll11 and Jag1 sending ability in *cis*, consistent with the mutual inhibition model of *cis* interactions.

## Lfng and Mfng modulate Dll1-Notch1 and Jag1-Notch1 *cis* interaction strengths differently

We next asked whether and how Fringe proteins modulate *cis* interactions. We constructed stable Notch1+Dll1 and Notch1+Jag1 cell lines constitutively expressing Lfng, Mfng, or Rfng. These constructs increased the *trans* response to plate-bound Dll1 ligands and decreased the response to Jag1 (in the case of Lfng or Mfng), or increased the response to both ligands (in the case of Rfng) consistent with previous results (***Hicks et al., 2000***).

To analyze the effect of Fringe proteins on *cis* interactions, we compared Notch availability in these cell lines to parental cell lines that lack ectopic Fringe expression. For the Notch1+Dll1 cell lines expressing Lfng, Mfng, or Rfng, Notch1 availability decreased to background levels in response to increasing Dll1 expression (***Figure 4A***). Note that absolute levels of fluorescence in the assay increase with Fringe expression, as can be observed at low total ligand levels in ***Figure 4A*** (inset), because Fringes enhance binding of the Dll1-Fc detection reagent to available Notch1. To account for this change in binding, we normalized the curves to the level of Notch availability measured when ligand levels were uninduced. After normalizing for this change, cell lines with Fringe expression appear to have stronger *cis* interactions (***Figure 4A***). Consistent with this, available Dll1 ligand assays revealed reduced available Dll1 when any of the Fringes was expressed (***Figure 4B***). Together, these results suggest that all three Fringe proteins preserve, or strengthen, Notch1-Dll1 *cis* interactions.

Fringe proteins had markedly different effects on Notch1-Jag1 *cis* interactions. In cell lines expressing Lfng or Mfng, even saturating expression of Jag1 was not sufficient to reduce Notch1 availability to background levels, in contrast to Dll1, indicating that Lfng and Mfng reduce *cis* interactions for Jag1, but not Dll1 (***Figure 4C***). In contrast, Rfng still allowed Notch1 availability to reach background levels at high *cis*-Jag1 levels. Note that here, as in ***Figure 4A***, we observed an increased binding of the Dll1-Fc detection reagent in cell lines expressing Fringe proteins. However, when Lfng or Mfng is expressed, only Dll1, and not Jag1, is able to reduce Notch availability to background levels. In the ligand availability assay, we did not detect a significant increase in Jag1 availability due to Lfng or Mfng

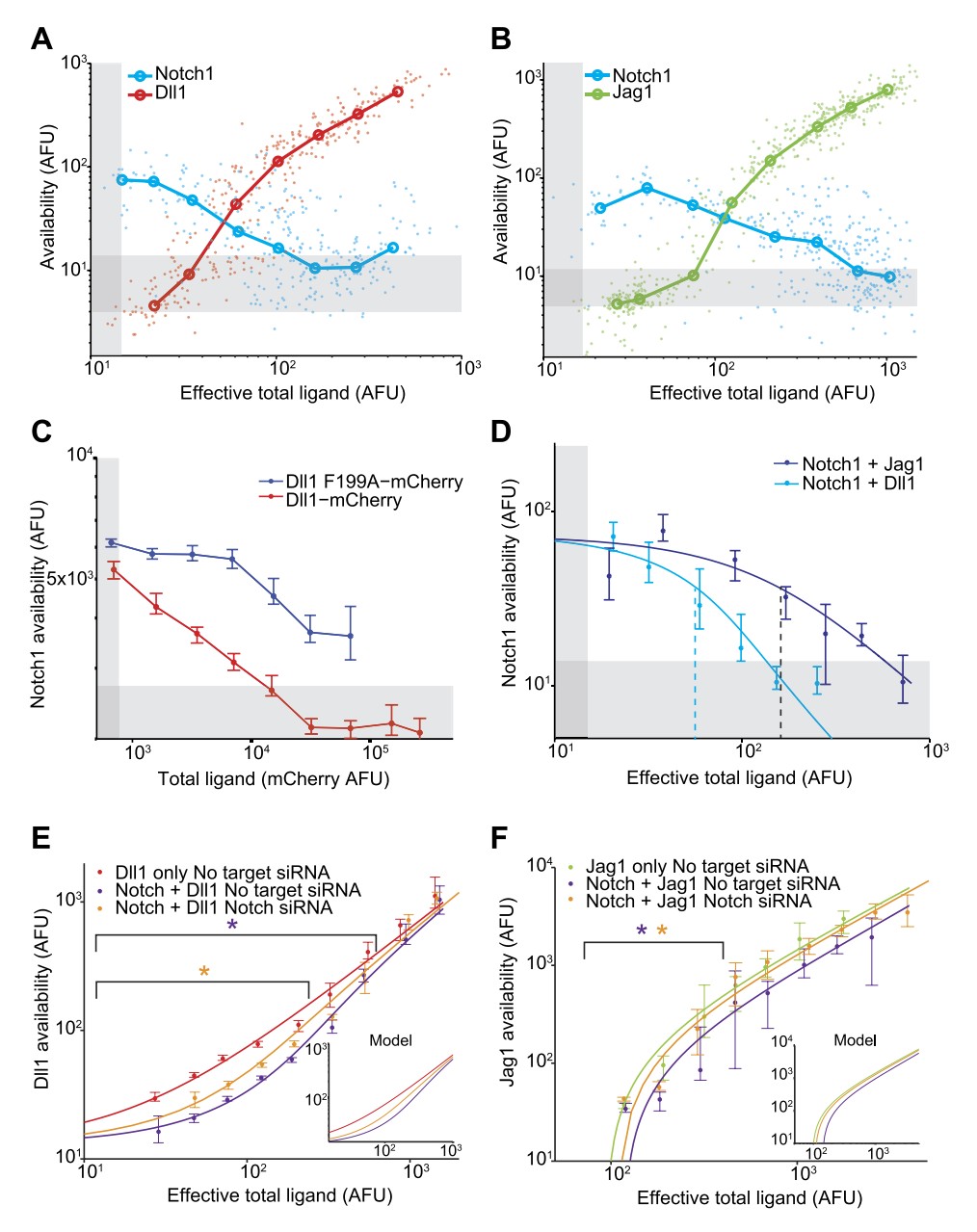

**Figure 3**. Dll1 and Jag1 exhibit mutual *cis*-inhibition with Notch1. (**A** and **B**) Single cell data show decreasing receptor availability and increasing ligand availability with increasing ligand expression. Circles denote the medians of data points in logarithmically spaced bins along the *x*-axis. 'Effective total ligand' refers to the ligand availability observed at a given ligand-CFP fluorescence value in a cell line expressing only ligand. For receptor availability data in **A**, n = 299 and in **B**, n = 352. For the ligand availability data in **A**, n = 323 and in **B**, n = 530. Gray bars in all panels represent background levels, defined as the 25–75 percentile range of fluorescence from parental CHO-K1 cells that do not express Notch1 or ligand-Cerulean constructs. (**C**) Transient expression of wild-type Dll1-mCherry (n = 8817), but not the Dll1-mCherry F199A mutant (n = 14,292), reduced Notch availability to background levels in a Notch1 cell line. Cells were analyzed by flow cytometry. Error bars in all panels denote 95% confidence interval for the bootstrapped estimate of the median. (**D**) Comparison of Notch availability in the Notch1+Dll1 and Notch1+Jag1 cell lines. Lines are fits to a model of receptor-ligand *cis*-interactions (Supplementary). (**E**) Comparison of ligand availability in cell lines expressing Dll1. Ligand availability a cell line expressing Dll1 only (n = 1146). Notch1 reduces ligand availability (purple, n = 1131), and this effect is rescued by siRNA against Notch1 (orange, n = 972). In the purple starred region, cells differ significantly in ligand availability between Notch1 or no target siRNA samples,

*Figure 3. Continued on next page*

*Figure 3. Continued*

while the orange star denotes regions where Dll1 and Notch1+Dll1 cells transfected with no target siRNA differ significantly. Significance was determined by applying the Wilcoxon rank sum test. Inset shows the model behavior for parameters derived from the fit in **D**. Knockdown of Notch was measured to be 50%. (**F**) Comparisons of ligand availability in a cell line expressing Jag1 only (green, n = 733), Notch1+Jag1 (orange, n = 532), and Notch1+Jag1 with siRNA against Notch (purple, n = 1163). Starred regions indicate significance as in **E**. Inset shows model behavior using parameters measured in **D**. Knockdown of Notch was measured to be 70%.

The following figure supplements are available for figure 3:

**Figure supplement 1**. Calibration of availability assay reagents.

**Figure supplement 2**. Receptor and ligand availability correlate with signal receiving and sending ability, respectively.

**Figure supplement 3**. Availability protocol and analysis pipeline.

**Figure supplement 4**. Untagged DSL ligands show similar effects to their fluorescently tagged counterparts.

expression (*Figure 4D*). Because the basal Jag1-Notch1 *cis*-inhibition strength is already weak compared to the Dll1-Notch1 *cis* interaction strength (*Figure 3D,F*), the Jag1 availability assay may not be sensitive enough to detect further reductions in *cis*-inhibition.

Together, these results suggest that Fringe expression modulates *cis*-inhibition between Notch1 receptor and ligands. Lfng and Mfng preserve and possibly strengthen interactions between Notch1 and Dll1, in *cis* and in *trans*, and weaken interactions between Notch1 and Jag1, in *cis* and in *trans*, while Rfng preserves or enhances the *cis* interactions with both ligands. Thus, the effects of Fringe proteins on *cis* interactions are in the same direction (strengthening or weakening) as their effects on *trans* interactions.

## Lfng enables cells to receive from *trans* Dll1 ligands, at high *cis* Jag1 levels

Because Lfng or Mfng expression weakens Notch1-Jag1 *cis* interactions, a Lfng-expressing cell could maintain high expression levels of *cis*-Jag1 without compromising its ability to receive signals from *trans* Dll1 ligands through Notch1. To test this prediction, we used a previously developed time-lapse video assay to titrate ligand levels over time in individual cells (*Figure 5A*) (*Sprinzak et al., 2010*). We constructed cell lines constitutively expressing the diverted chimeric receptor, hN1ΔICD-Gal4esn, and incorporating a UAS-H2B-Citrine reporter activated by Gal4 released by activated chimeric Notch1 (*Figure 5B*). To these cell lines we added a tetracycline-inducible Jag1-mCherry ligand. Finally, to analyze the effect of Lfng on Notch1-Jag1 *cis*-inhibition, we added to this parental cell line a stably integrated, constitutively expressed Lfng gene. We also constructed similar cell lines with Dll1-mCherry in place of Jag1-mCherry.

Both parental and Lfng-expressing cell lines were seeded on plates coated with Dll1ext-Fc recombinant protein to activate Notch1 in *trans*, but signaling was initially blocked with the Notch signaling inhibitor DAPT. 24 hr before the start of the video (*t* = −24 hr), we added 4-epiTc to induce *cis*-Jag1 production. 1 day later, at a time defined as *t* = 0, we washed out 4-epiTc and DAPT, halting further *cis*-Jag1-mCherry production, and allowing Notch signaling to commence. We then monitored the cells for 2 days using time-lapse fluorescent microscopy. As the cells divided, *cis*-Jag1 levels gradually diminished in individual cells, predominantly through dilution (*Figure 5C*). To measure the dependence of Notch activation on *cis* ligand levels, we used custom image analysis software to quantify fluorescence in individual cells in the video ('Materials and methods'). We then plotted the rate of increase of H2B-Citrine, a measure of Notch activity, against mCherry fluorescence, a measure of total ligand abundance, for each cell.

In the parental cell lines Notch reporter activation was delayed by ~24 hr, indicative of *cis*-inhibition (*Figure 5D*, *Video 1*). By contrast, the Lfng cell line responded earlier to the plate-bound Dll1ext-Fc despite high *cis*-Jag1 levels, and not significantly later than in cells lacking ligand expression altogether, indicating that Lfng reduces Notch1-Jag1 *cis* interaction, and thereby prevents high *cis*-Jag1

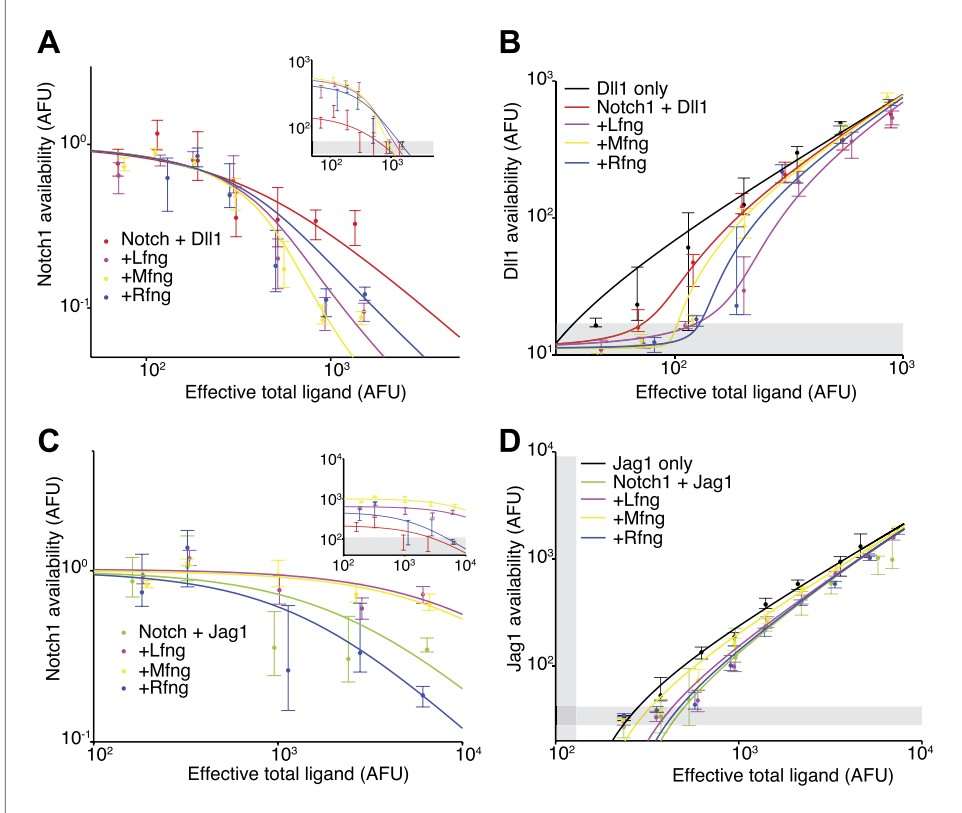

**Figure 4**. Fringe proteins show distinct effects on Jag1-Notch1 and Dll1-Notch1 *cis* interactions. (**A**) Available Notch1 levels for the Notch1+Dll1 cell line without Fringe (red) or with Lfng (magenta), Mfng (orange), or Rfng (blue). Lines are fits to model ('Materials and methods'). Addition of any of the three Fringes accelerates the drop-off of Notch1 availability. In the inset, the same data, but unnormalized, shows that addition of any of the three Fringe proteins does not prevent available Notch1 from reaching background levels. (**B**) Dll1 availability for the cell lines from **A**. (**C**) Similar to **A**, but for the Notch1+Jag1 cell lines. Addition of Lfng and Mfng prevents the depletion of Notch1 availability, while addition of Rfng accelerates depletion of Notch1 availability. In the inset, the unnormalized data shows that Lfng or Mfng, but not Rfng, can block the ability of Jag1 to reduce Notch1 availability to background levels. (**D**) Jag1 availability for the cell lines in **C**. In all panels, points represent medians of data points in evenly spaced bins taken along the log of the x-axis. Error bars are the 95% confidence intervals of the bootstrapped estimated of the bin medians. Solid lines are model fits to the single-cell data (see 'Materials and methods'). Gray bars denote the 10–90th percentile fluorescence range of stained parental CHO-K1 cells that do not express Notch1 or ligands.

levels from blocking Notch activation (*Figure 5C,D*, *Video 2*). In contrast, when we performed the same experiment with cell lines expressing Dll1-mCherry, we observed no corresponding relief of *cis*-inhibition due to Lfng, suggesting that Lfng does not weaken Dll1-Notch1 *cis* interactions. These results, consistent with those from the availability assay, support the finding that Lfng inhibits Jag1-Notch1, but not Dll1-Notch1, *cis*-inhibition.

## Fringe differentially regulates Delta-Notch and Serrate-Notch *cis* interactions in the *Drosophila* wing

To determine whether Fringe modulation of *cis* interactions occurs in a developmental context, we turned to *Drosophila* wing imaginal disc as a model system. Although the Notch pathway has fewer components in flies than vertebrates, the *Drosophila* wing disc provides a tractable system to examine related behaviors in a developmental context. Moreover, previous work has established both an effect of Fringe on *trans* interactions that is qualitatively similar to the mammalian Lfng and Mfng, as well as a clear role for *cis*-inhibition of Notch by its ligands Delta and Serrate (related to mammalian Jag1) in this process (*Doherty et al., 1996*; *Micchelli et al., 1997*; *Panin et al., 1997*).

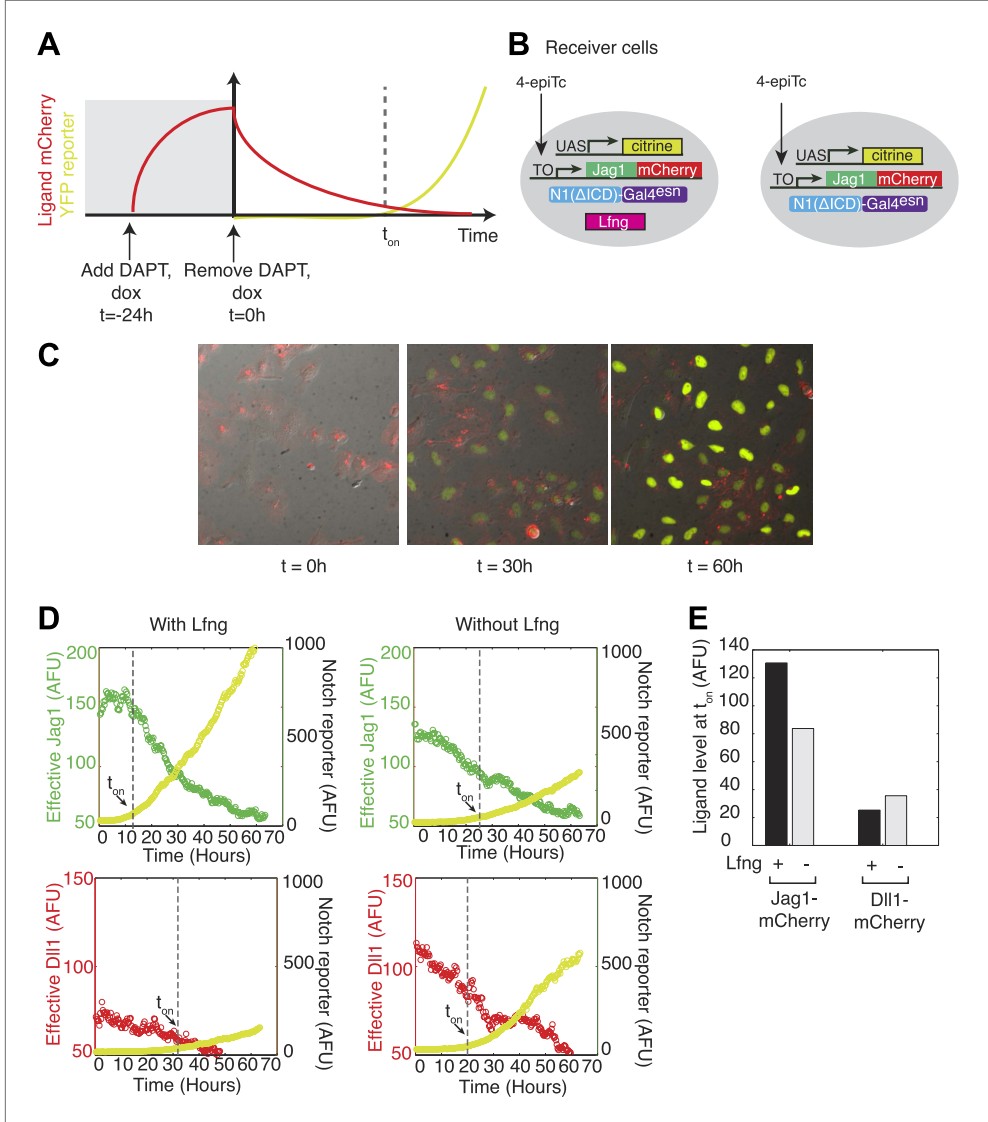

**Figure 5**. Lfng relieves Jag1-Notch1 but not Dll1-Notch1 *cis*-inhibition. (**A**) Schematic design of time-lapse experiment to analyze *cis*-interactions. Before the video, cells are pre-induced to express high levels of ligand and then seeded on plates coated with Dll1-Fc. During the video, cell division dilutes the *cis*-ligand sufficiently to allow cells to respond to plate-bound ligands. (**B**) Cell lines expressing 'diverted' Notch1-Gal4 chimeric receptor, a fluorescent reporter for Notch signaling (UAS-H2B-citrine), and inducible Jag1-mCherry ligand, with (left) and without Lfng (right). Corresponding cell lines with the Dll1-mCherry ligand are not shown. (**C**) Typical video filmstrip. (**D**) Quantification of videos of the Notch1+Jag1 and Notch1+Dll1 cell lines, with and without Lfng. Points show the mean fluorescence of all cells in a single frame. The cell line with Lfng responds earlier than the cell line without Lfng, reflecting a weaker *cis* interaction between Jag1 and Notch1. The time when the YFP slope exceeds a threshold, defined as 10% of the final YFP slope, is marked as $t_{on}$. (**E**) Quantification of the ligand levels at $t_{on}$ for each cell line. Values are the average of two videos. Notch activity occurs even at high *cis*-Jag1 levels in the +Lfng, but not the −Lfng, cell line. Notch responses occurred only at low ligand levels for the Dll1-mCherry cell lines, with and without Lfng.

The adult *Drosophila* wing has a smooth margin lined with bristles and is patterned with five wing veins (***Figure 6A***). Normal development of both the margin and the wing veins relies on spatially restricted Notch signaling. In the case of the wing margin, a sharp stripe of Notch signaling occurs at the boundary between dorsal wing cells that express Notch, Fringe, and the Delta and Serrate ligands, and ventral wing cells that express Notch and Delta. This Notch signaling drives the expression of the

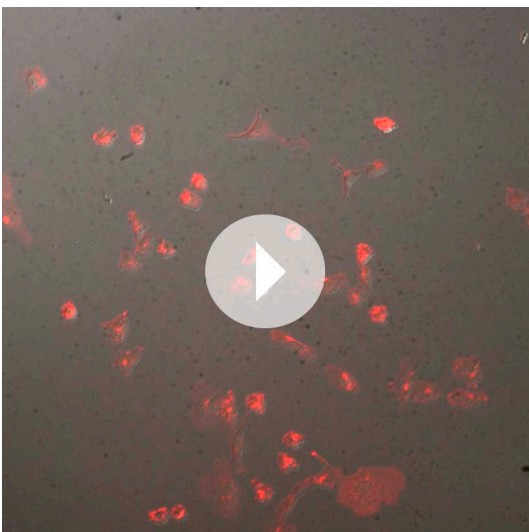

**Video 1**. Dilution video assay with a cell line expressing Notch1+Jag1. Reporter activation shows a delay, as cell divisions are required to dilute out *cis*-Jag1. Frame rate is 10 fps, with a frame taken ever 20 min.

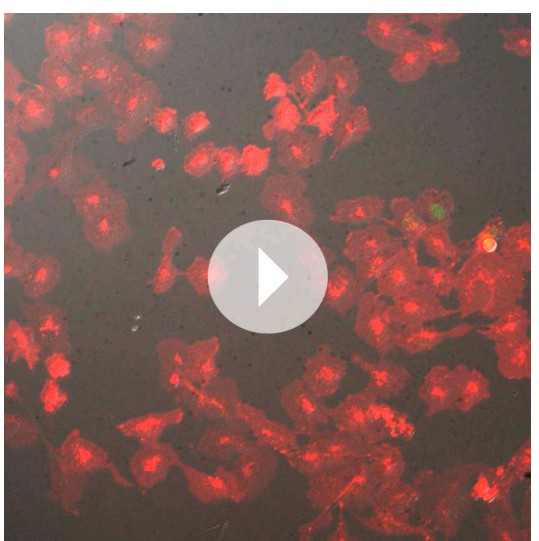

**Video 2**. Dilution video assay with a cell line expressing Notch1+Jag1+Lfng. Reporter activation is immediate despite high *cis*-Jag1 levels. Frame rate is 10 fps, with a frame taken ever 20 min.

downstream targets E(spl), Wingless, and Cut, that proceed to further activate the expression of ligands and to direct the patterning and development of the wing margin (*de Celis and Bray, 1997*). The wing veins are specified by a gradient of Delta ligand expression that drives Notch signaling in two stripes of cells that form the edges of the wing veins (*de Celis et al., 1997*; *De Celis, 2003*).

Removing one copy of *Notch* results in wing vein thickening and the classic 'notched' wing phenotypes (*Figure 6B*) (*Mohr, 1919*). If Notch and its ligands were solely involved in *trans*-activation, one would expect the *Notch* haploinsufficient phenotypes to be enhanced by a simultaneous decrease in ligand expression. However, loss of one copy of *Delta* was reported to suppress the $Notch^{+/-}$ wing vein phenotype (*de Celis and Bray, 2000*), indicating that the haploinsufficient phenotypes observed in $Notch^{+/-}$ animals are caused by *cis*-inhibition of Notch due to an increase in the relative levels of ligands to Notch.

The cell-based assays indicated that Lfng and Mfng preserve the *cis*-inhibition of Notch by Dll1 but weaken the *cis*-inhibition of Notch by Jag1. To examine whether *Drosophila* Fringe affects the ability of Delta and Serrate to *cis*-inhibit Notch in a manner similar to Lfng and Mfng, we performed genetic interaction studies in a $Notch^{+/-}$ background, which seems to be more sensitive to *cis*-inhibition by ligands. Because the wing margin loss phenotype is only partially penetrant in $Notch^{+/-}$ animals, we used a classification system to quantify this phenotype in various genotypes (*Figure 6L*): A wild-type wing is scored as 0, a wing with mild margin loss adjacent to the L3 wing vein is scored as 1, a wing with more extensive wing margin loss extending to L3 and L4 wing vein regions is scored as 2, and a wing with margin loss in L3-L4 and anterior regions of the wing is scored as 3. Based on this scoring system, only 40% of $Notch^{+/-}$ wings show a margin defect, all of them with a score of 1 (*Figure 6M*).

First, we sought to examine the effects of increasing ligand gene dosage on *Notch* haploinsufficient phenotypes. To this end, we first generated transgenic flies harboring the *Delta* locus ($Dl^{gt-wt}$) or the *Serrate* locus ($Ser^{gt-wt}$), in which the expression of Delta or Serrate is driven by endogenous promoter/enhancers, and showed that these transgenes behave similarly to their endogenous counterparts in flies (*Figure 6—figure supplement 1*). Animals with one or two copies of the $Dl^{gt-wt}$ transgene in a wild-type background did not exhibit any wing defects (*Figure 6—figure supplement 1C* and not shown). Similar effects have been observed in $Dp(3;3)bxd^{110}/+$ flies (not shown), which harbor one copy of a homozygous lethal intrachromosomal duplication containing the *Delta* locus (*Vassin and Campos-Ortega, 1987*). However, in $Notch^{+/-}$ animals, one copy of the $Dl^{gt-wt}$ transgene resulted in a slight increase in the penetrance of class 1 wing margin defects and a moderate enhancement of the wing vein thickening (*Figure 6C,M*). Two copies of $Dl^{gt-wt}$ strongly enhanced

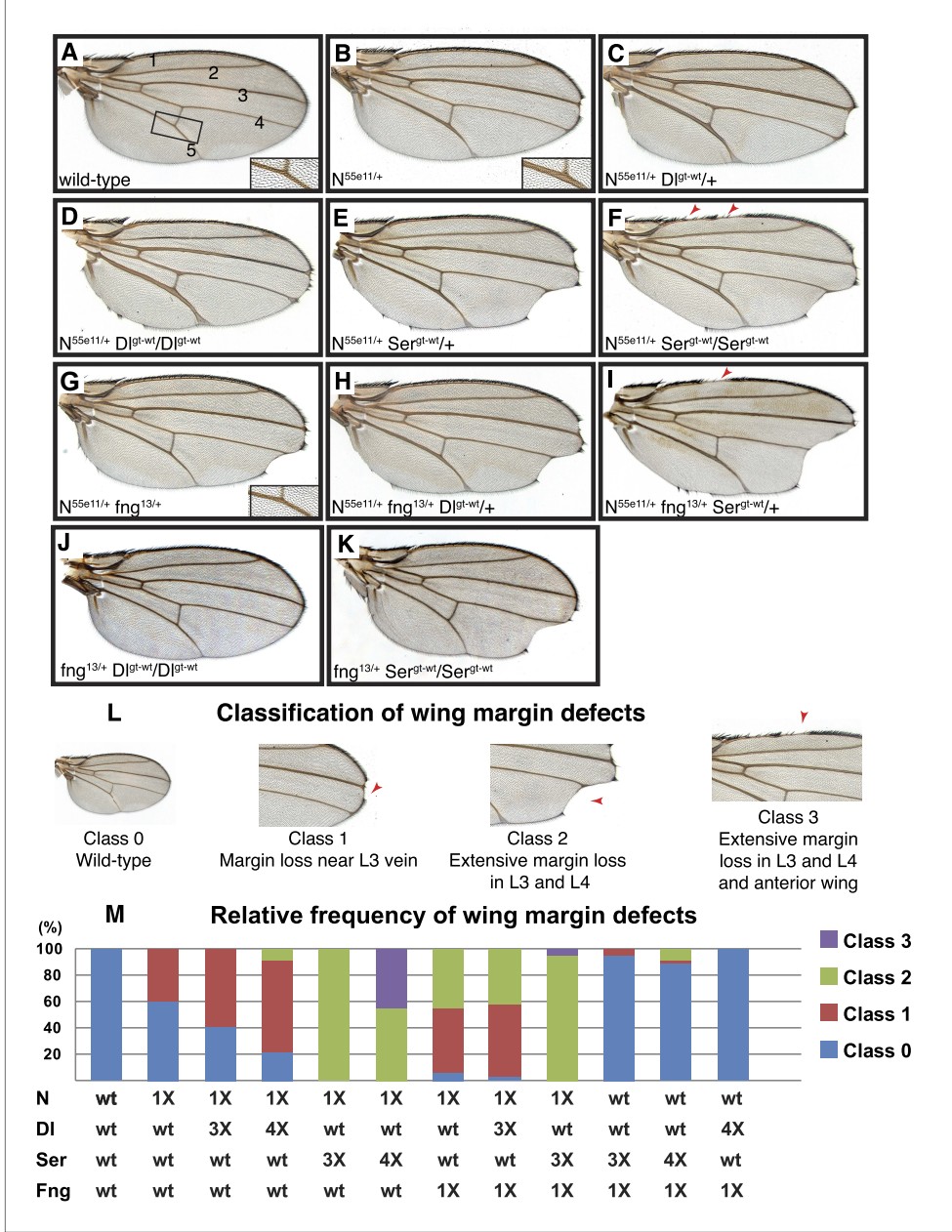

**Figure 6**. Delta and Serrate have distinct *cis*-inhibition phenotypes, and decreasing Fringe activity affects each of these phenotypes differently. (**A**) A wild-type fly wing. The five wing veins are marked. Inset is a close-up of the area marked by the rectangle. (**B**) Loss of one copy of *Notch* leads to mild wing margin loss and wing vein thickening (inset). (**C** and **D**) One (**C**) or two (**D**) additional copies of *Delta* in the $N^{55e11/+}$ background from **B** leads to mild enhancement of the wing margin defects and strong enhancement of vein thickening phenotypes. (**E** and **F**) One (**E**) or two (**F**) additional copies of *Serrate* in the $N^{55e11/+}$ background from **B** results in strong enhancement of wing margin defects but only mild enhancement of vein thickening phenotypes. (**G**) Loss of one copy of *fringe* ($fng^{13/+}$) in the $N^{55e11/+}$ background from **B** enhances wing margin loss. (**H**) Addition of one copy of *Delta* to the $N^{55e11/+}$ $fng^{13/+}$ background from **G** does not further enhance margin loss and seems to suppress the vein thickening phenotype (compare to **C**). (**I**) Addition of one copy of *Serrate* to the $N^{55e11/+}$ $fng^{13/+}$ background further enhances wing margin loss in **G**, suggesting that Fringe usually works to counter the effects of Serrate overexpression. (**J**) Loss of one copy of *fringe* in a Delta overexpression background does not lead to any wing margin defects. (**K**) Loss of one copy of Fringe in a Serrate overexpression background leads to wing margin loss in some animals, suggesting that Fringe normally blocks the negative effects of Serrate on Notch signaling in animals with wild-type Notch expression levels. (**L**) Classification system used to quantify frequencies of mutant phenotypes. Class 0 denotes a wild-type fly

*Figure 6. Continued on next page*

*Figure 6. Continued*

wing morphology. Class 1 flies show mild wing margin loss adjacent to the L3 vein. Class 2 flies show more extensive margin loss extending to the L3 and L4 veins. Class 3 flies show margin loss in L3 and L4 and also in anterior regions of the wing. (**M**) Quantification of the phenotypes using the scoring system in **J**. At least 50 wings were scored for each genotype, except for the last two columns, for which we scored 48 and 34 wings, respectively. The most severe class 3 defects arise when *Notch* dosage is halved (1X Notch) and *Serrate* dosage is doubled (4X Serrate), a consequence of Serrate *cis*-inhibition. Class 3 defects also arise when *Notch* and *fringe* dosages are halved, and an additional copy of *Serrate* is added, suggesting that Fringe normally works to block the effects of Serrate *cis*-inhibition (Compare column 5 with column 9).
The following figure supplement is available for figure 6:

**Figure supplement 1**. Genomic Delta and Serrate transgenes behave similarly to endogenous copies of Notch ligands in Drosophila.

---

the wing vein thickening phenotype in *Notch*[+/−] animals (***Figure 6D***). These animals also showed a moderate enhancement of the wing margin loss phenotype (***Figure 6D,M***). These observations indicate that although Delta-Notch *cis*-inhibition affects both wing vein and wing margin formation, Notch is more sensitive to Delta *cis*-inhibition during wing vein formation.

The effects of increasing the gene dosage of *Serrate* on the *Notch* haploinsufficient phenotypes were quite different from those of increasing *Delta* dosage. Two copies of the *Ser*[gt-wt] transgene did not cause any wing abnormalities in a wild-type background (***Figure 6—figure supplement 1D***). However, one copy of *Ser*[gt-wt] significantly enhanced the wing margin loss phenotype of *Notch*[+/−] animals without affecting their wing vein phenotype (***Figure 6E***). A second copy of *Ser*[gt-wt] further enhanced the wing margin loss (***Figure 6F***). Indeed, 44% of the *Notch*[+/−]; *Ser*[gt-wt]/*Ser*[gt-wt] wings showed a Class 3 wing margin loss, which was not observed in any of the other genotypes analyzed in this study so far (***Figure 6M***). Of note, despite their severe wing margin loss, *Notch*[+/−]; *Ser*[gt-wt]/*Ser*[gt-wt] animals only showed a mild enhancement of the wing vein thickening compared to *Notch*[+/−] flies (***Figure 6F***, compare to ***Figure 6B***). These observations indicate that although both Delta and Serrate can *cis*-inhibit Notch during wing development, Notch is more sensitive to *cis*-Delta during wing vein formation and more sensitive to *cis*-Serrate during wing margin formation.

Next we tested whether and how altering Fringe activity changed these distinct Delta and Serrate *cis*-inhibition phenotypes. We used *fringe*[13] and *fringe*[L73] strains that harbor severe loss-of-function alleles with nonsense mutations in Fringe (***Correia et al., 2003***). Animals heterozygous for *fringe* have normal wings (***Correia et al., 2003***). However, loss of one copy of *fringe* alters the *Notch* haploinsufficient wing margin and wing vein phenotypes in opposite directions: it enhances the Serrate-dependent wing margin loss but suppresses the Delta-dependent wing vein thickening (***Figure 6G,M***). Addition of one copy of *Serrate*, but not one copy of *Dl*, enhances the wing margin loss in *Notch*[+/−]; *fng*[+/−] animals (***Figure 6H,I,M***). Indeed, even in *fng*[+/−] animals with wild-type *Notch* gene dosage, addition of a copy of *Serrate* results in a low penetrance Class 1 wing margin loss (***Figure 6M***) and addition of two copies of *Serrate* enhances this phenotype by generating Class 2 wing margin loss in ~8% of wings (***Figure 6K,M***). Of note, adding one or two copies of *Delta* does not result in any wing phenotypes in a *fng*[+/−] background (***Figure 6J,M*** and data not shown). If Fringe only affects the *trans* activation of Notch by ligands, one would expect to see increased signaling and therefore wing margin duplication in *fng*[+/−] animals with additional copies of *Serrate*. However, not only do we not see an enhancement of Notch signaling in these animals, we also observe a wing margin loss which becomes more severe as we increase *Serrate* gene dosage. This indicates that in addition to its effect on Notch *trans* activation by Serrate, Fringe also regulates the *cis*-inhibition of Notch by Serrate. Altogether, these observations strongly support the conclusion that during *Drosophila* wing development, Fringe increases the sensitivity of Notch to *cis*-Delta but decreases its sensitivity to *cis*-Serrate, similar to the effect of Lfng and Mfng in the mammalian context.

## Discussion

Interactions between Notch ligands and receptors, both in *cis* and in *trans*, control the quantitative ability of cells to send or receive signals. Recent work with cells expressing one ligand (Dll1) and one receptor (Notch1) showed that strong and mutually inactivating *cis* interactions between Dll1 and

Notch1 can force cells into mutually exclusive signaling states (*Figure 1A*) (*Sprinzak et al., 2010*). But how does this property persist or change in more complex contexts involving multiple ligand species and modulation by Fringe proteins? More generally, how do *cis* and *trans* interactions, together with Fringe proteins, specify a particular set of signaling states, and how do those states determine the directionality and specificity of signaling?

Extrapolating from the data presented here, one can map out the set of possible signaling states generated by expression of various combinations of Notch1, Dll1 or Jag1 ligands, and one of the three mammalian Fringes. To identify the most qualitatively distinct possible signaling states, we analyzed the limiting cases of extremely high or low receptor/ligand expression ratios at either very low or very high Fringe levels, keeping in mind that the quantitative sending and receiving capabilities of real cells will generally depend on the actual expression levels.

For the Notch1+Dll1 cells, all three Fringe proteins maintained or strengthened mutually inhibitory *cis* interactions between Dll1 and Notch1. Thus, with or without Fringe expression, the ability of cells to send using Dll1 depends on the relative levels of Notch1 and Dll1 (*Figure 7A,B*). Rfng preserves or strengthens Notch1 interactions with both Dll1 and Jag1, in *cis* and *trans*. As a result, Rfng expression maintains the same qualitative send or receive signaling states as observed without Rfng (*Figure 7B,C*, second and third panels). On the other hand, Lfng and Mfng preserve or strengthen Notch1-Dll1 *cis* and *trans* interactions, but diminish Notch1-Jag1 *cis* and *trans* interactions (*Hicks et al., 2000*; *Ladi et al., 2005*). Consequently, their expression can favor receiving from Dll1 over Jag1 (*Figure 7C*, first panel), without affecting the sending state (*Figure 7C*, first and third panels). Thus, for a cell expressing Notch1 and Dll1, expression of Fringe proteins does not appear to disrupt the exclusivity of sending and receiving capabilities.

The picture is quite different, however, for Notch1+Jag1 cells (*Figure 7D,E*). Without Fringe, these cells can exhibit exclusive send and receive states similar to cells expressing Notch1 and Dll1 (*Figure 7D*). As with Dll1, Rfng enhances *cis* and *trans* Notch1-Jag1 interactions (*Figure 7D*, first and second panels), maintaining exclusive sending and receiving signaling states. However, expression of Lfng or Mfng produces a qualitatively different behavior. By weakening Jag1-Notch1 *cis* interactions, they enable cells to simultaneously maintain Notch1 and Jag1 availability on the cell surface. A cell in this state could then use Notch1 to receive from *trans* Dll1 ligands (but not from Jag1, as Lfng/Mfng block Jag1-Notch1 *trans* signaling), while also activating other cells with the Jag1 ligand (*Figure 7E*, fourth panel). Thus, a cell expressing Lfng/Mfng, Jag1, and Notch1 has the interesting property of being able to send and receive simultaneously, *but only using different ligands*.

In this case, the Notch pathway allows for simultaneous sending and receiving by a single cell, but not efficient signaling among a set of identical cells in this signaling state. This analysis suggests the intriguing possibility that the Notch pathway architecture might generally be structured to favor heterotypic signaling (between cells in different states) over homotypic signaling (between cells in the same state). This can be seen in the schematic representations of *Figure 7*, where no cell contains both an incoming and outgoing arrow on the same ligand (same color), indicating that cells in each of these states should not be able to signal efficiently to other cells in the same state. Whether Notch signaling is broadly heterotypic across all contexts remains to be tested.

## Signaling states in fly wing disc dorsal-ventral boundary formation

The signaling states shown in *Figure 7* can provide insights into developmental processes. For example, consider boundary formation in the *Drosophila* wing imaginal disc, a well-characterized developmental process involving both Notch ligands and Fringe (*Micchelli et al., 1997*; *Panin et al., 1997*; *de Celis and Bray, 1997*; *Irvine, 1999*; *Troost and Klein, 2012*). In the first stage of boundary formation, cells within the dorsal and ventral compartments of the wing disc signal to one another, forming a sharp stripe of Notch activation at the interface between the two cell populations. This Notch activity drives the expression of the Notch target genes E(spl), Wingless, and Cut, which together refine the expression patterns of Notch ligands and regulate the spatial pattern of Notch activation in subsequent stages (*de Celis and Bray, 1997*). For this discussion, we focus on the initial step before these feedback mechanisms become active. Dorsal cells express Notch, Delta, Serrate, and Fringe, while ventral cells express Notch and Delta only (*Figure 8*). Because Fringe blocks Serrate-Notch *trans* signaling, dorsal cells do not signal to one another with Serrate, but can signal with Serrate to ventral cells (*Irvine, 1999*). However, Fringe also promotes Delta-Notch *trans* signaling, so dorsal cells respond to Delta expressed by ventral cells. In this way, signaling can only occur at the interface between the two cell populations (*Fortini, 2009*).

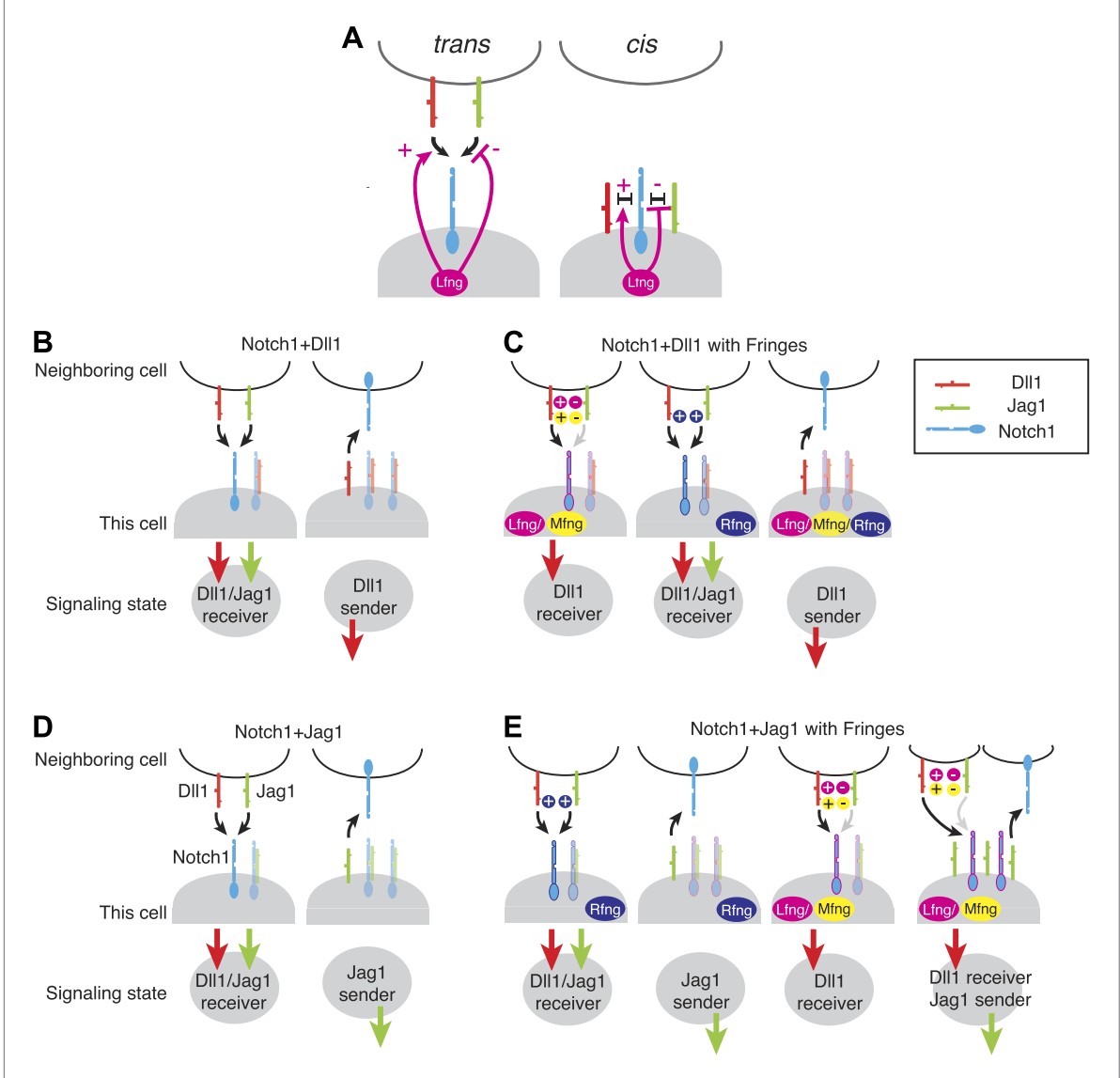

**Figure 7**. Fringe modulation of *cis* interactions generates a set of distinct Notch signaling states. (**A**) Lfng or Mfng modification of Notch1 enhances *trans* activation from Dll1 and weakens trans activation from Jag1 (left). Lfng or Mfng modification of Notch1 enhances *cis* interactions with Dll1 and weakens cis interactions with Jag1. In (**B**–**E**), each cartoon denotes a distinct configuration of Notch pathway components, with the resulting signaling state indicated schematically below. We consider extreme endpoints, where Notch1 expression is much higher than ligand expression (Notch1 > ligand, light shaded panels), and where ligand expression is much higher than Notch1 expression (Ligand > Notch1, dark shaded panels). We also consider either low (**A** and **C**) or very high levels of Fringe expression (**B** and **D**). (**A**) A cell expressing Notch1 and Dll1 can be in a receiving state, where it can be activated by *trans* Dll1 or Jag1, when Notch1 levels surpass Dll1 levels, left. A Dll1-sending state occurs when Dll1 exceeds Notch1, right. (**B**) With the addition of Lfng or Mfng, the receiving state in **A** becomes sensitive to *trans* Dll1 but not *trans* Jag1 (left). Rfng enhances receiving from both ligands (middle). Any of the three Fringe proteins support the Dll1 sending state when Dll1 exceeds Notch1 (right). (**C**) Co-expression of Notch1 and Jag1 permits exclusive sending (right) or receiving (left) signaling states, similar to those in **A**. (**D**) Cells expressing Notch1, Jag1, and Rfng show exclusive send or receive signaling states as in **A** and **C** (first two panels). However, addition of Lfng or Mfng inhibits Notch1-Jag1 *cis* interactions. As a result, these cells can receive signals from Dll1 but not Jag1 when Notch1 expression exceeds Jag1 expression (third panel). Finally, when Jag1 exceeds Notch1, the cell can send with Jag1 and receive from Dll1 simultaneously (right panel). (**E**) The possible signaling states of the Notch pathway. Receptor ligand interactions prohibit cells from sending to themselves (no self arrows) and also disallow cycles.

This picture of boundary formation appeared to challenge the notion that send and receive signaling states are exclusive (*Figure 1A*). Indeed, elegant mosaic experiments based on mis-expression of ligands (*de Celis and Bray, 1997*), have suggested that both dorsal and ventral cells are capable of

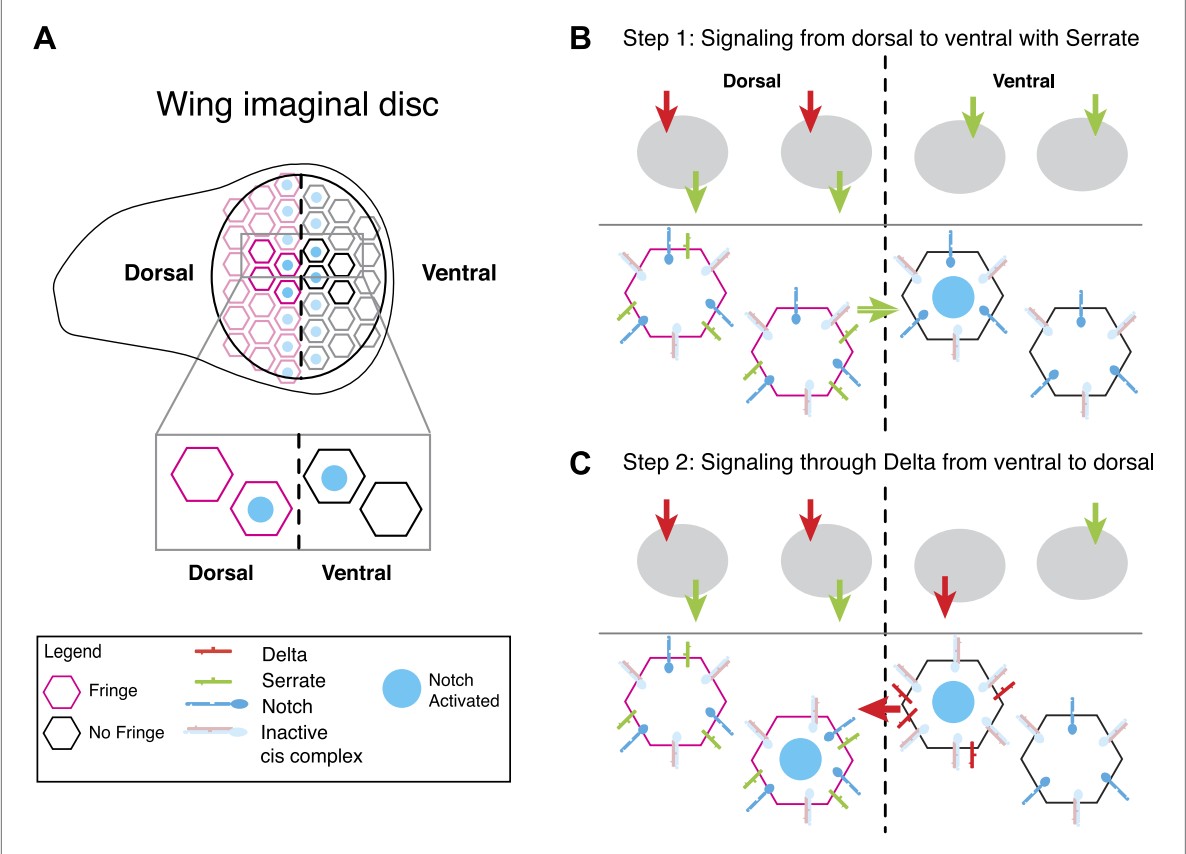

**Figure 8**. A model for Notch signaling states during dorsal-ventral boundary formation in the Drosophila wing disc. (**A**) A schematic of the Drosophila wing imaginal disc during the third larval instar. During this stage of development, a sharp stripe of Notch signaling occurs at the interface of the dorsal and ventral wing compartments (blue nuclei), leading to upregulation of target genes that drive further wing development. (**B**) The first step of boundary formation, with signaling states indicated in cartoons above. Initially, ventral cells express Notch and Delta. Because ventral cells can receive signal, Notch must be in excess of Delta to achieve a receiving signaling state. Dorsal cells express Serrate, Notch, Delta, and Fringe (magenta outline denotes Fringe expression). Fringe promotes Delta-Notch *cis* interactions but weakens Serrate-Notch *cis* interactions, enabling dorsal cells to simultaneously receive signals from Delta while sending signals with Serrate. Thus, the first signaling step occurs from dorsal to ventral cells (green arrow). We cannot rule out that there may be some low level signaling from Delta expressed in dorsal cells as suggested by *Lei et al. (2003)*; however, experiments suggest that Delta in ventral cells is dispensable for proper wing development. (**C**) The second step of signaling begins with upregulation of Delta in activated ventral cells. These cells switch to a sending state, and *trans* activate dorsal cells. We cannot rule out the possibility that these cells send signal to their ventral cell neighbors; however, because Delta cannot efficiently send in a Fringe-negative background, and because of the existence of feedback mechanisms at this stage, Notch activation is kept to low enough levels to prevent upregulation of target genes involved in wing margin formation (*de Celis and Bray, 1997*). The result is the observed pattern of Notch activation at the boundary of dorsal and ventral compartments.

receiving signals, that is, all cells are in a receiving state. But in order for signaling to occur, some cells must also possess the capability to send signals. Thus, it appears that some cells should be able to send and receive signal simultaneously.

Fringe modulation of *cis* interactions, reported above, together with a recent developmental study of this system (*Troost and Klein, 2012*) suggest a potential resolution to this discrepancy. *Troost and Klein (2012)* recently examined signaling at the boundary with higher time resolution than in previous work. They found that in the early stages of wing margin formation, signaling occurred in two sequential phases: first, dorsal cells signal via Serrate to ventral cells. Subsequently, ventral cells up-regulate Delta and signal back to dorsal cells.

In this model, because ventral cells are initially capable of receiving, they should have an excess of Notch over Delta, as shown in *Figure 8B* (ventral). The data presented here suggest that in dorsal cells, Fringe should strengthen *cis*-inhibition between Delta and Notch, while weakening *cis*-inhibition between Serrate and Notch. Because dorsal cells are initially in a receiving state (*de Celis and Bray,*

*1997*), Delta expressed by these cells should not efficiently activate Notch in neighboring dorsal cells. Indeed, *Delta* mutant clones in the dorsal compartment do not affect adult wing margin formation (*Doherty et al., 1996*), indicating that Delta-mediated signaling among dorsal cells, which occurs at most very weakly (*Lei et al., 2003*), is dispensable for normal wing margin formation. The formation of a mutually inactivating, *cis*-inhibitory interaction between Delta and Notch could explain why dorsal cells do not strongly activate one another even though they express and are responsive to Delta (*Doherty et al., 1996*; *Lei et al., 2003*). At the same time, Fringe could allow Serrate and Notch to both remain available simultaneously (*Figure 8B*, dorsal), a state resembling that in *Figure 7E* (far right panel). Thus, in the first phase, signaling would occur only from dorsal to ventral, as observed (*Troost and Klein, 2012*). It is worth mentioning that the dynamic and complex expression patterns of Notch, ligands and Fringe in developmental contexts could superimpose additional layers of regulation on the effects of Fringe on *cis* and *trans* interactions between Notch and ligands.

Notch activation in ventral cells in the first phase (*Figure 8B*) causes them to up-regulate Delta expression, switching them to a Delta-sending state (*de Celis and Bray, 1997*) (*Figure 8C*). In this second phase, ventral cells can *trans*-activate dorsal cells, which remain responsive to Delta. In principle, ventral sender cells could send to adjacent ventral cells; however it seems that at least in flies, Delta cannot send efficiently to cells that lack Fringe expression, and therefore in the flanking ventral cells Notch activity cannot achieve high enough levels to induce Cut and Wingless expression (*de Celis and Bray, 1997*). This interpretation requires Fringe modulation of *cis* interactions, as observed here, and implies that signaling during boundary formation is heterotypic (*Troost and Klein, 2012*). It will be interesting to see if other Notch-dependent boundary formation processes involve similar Notch signaling states and dynamics (*Irvine, 1999*).

## Knowledge of Fringe effects on *cis* interactions could help resolve contradictory findings

These results could help explain other puzzling observations. For example, one of the most striking vertebrate Notch phenotypes is disorganized somitogenesis in Notch pathway mutants (*Irvine, 1999*). In mice and chicks, Lfng is required for this process (*Lewis, 1998*). However, Lfng inhibits Notch1-Dll1 signaling (*Dale et al., 2003*), rather than promoting signaling as expected from previous analysis of its effect on *trans* interactions (*Hicks et al., 2000*). The ability of Lfng to strengthen Dll1-Notch1 *cis* interactions could explain this phenomenon, since Lfng would tend to reduce the abundance of Dll1 and Notch1 available for *trans* signaling interactions.

Based on these results and others, different configurations of receptors and ligands, through *cis* interactions, could work to specify distinct signaling states in cells (*Figure 7B–E*). These states are more complex than just 'send' and 'receive' but are still highly constrained by *cis* interactions and the ways they can be modulated. Understanding what signaling states are possible, how expression levels of various pathway components determine those signaling states, and how cells in different signaling states interact with one another, could provide a useful way to think about the Notch signaling system more generally and to infer the directionality and specificity of signaling in more complex contexts.

However, many questions remain. We still lack systematic measurements of the interaction strengths, in *cis* and in *trans*, for the full repertoire of ligand–receptor pairs, and their quantitative dependence on Fringe expression levels. Given that multiple Fringe proteins are co-expressed in many systems, we will need to examine how Fringe proteins combine to influence Notch signaling. Fringe proteins are also known to modify the DSL ligands (*Panin et al., 2002*); however, a functional role for sugar modifications of the DSL ligands has not been forthcoming (*Muller et al., 2014*). A potential effect of these modifications on *cis* interactions, if it exists, would also need to be accounted for in a more complete model. Finally, additional components beyond ligands, receptors, and Fringes may need to be considered in developing a more predictive view of Notch signaling. We anticipate that the experimental approaches developed here can be generalized to address these questions and provide a deeper understanding of the basic design principles of the Notch signaling system.

## Materials and methods

### Cell culture

CHO-K1 cells were maintained as described in *Sprinzak et al. (2010)*. Briefly, cells were maintained in Alpha-MEM Earle's Salts media (Irvine Scientific, Santa Ana, CA) supplemented with 10% Tet-system

approved FBS (Clontech, Mountain View, CA), and L-glutamine, penicillin and streptomycin additive (Gibco, Carlsbad, CA), and stored in an incubator at 37°C at 5% $CO_2$.

## Transfection and cell line generation

Genetic constructs, including siRNA constructs, were introduced into cells using Lipofectamine 2000 reagent according to the manufacturer's protocol (Life Technologies, Carlsbad, CA), or FugeneHD reagent (Promega, Madison, WI). Selection was performed using 400 µg/ml Zeocin (Life Technologies), 10 µg/ml Blasticidin (InvivoGen, San Diego, CA), 600 µg/ml Geneticin (Life Technologies), 500 µg/ml Hygromycin (InvivoGen) and/or 3 µg/ml Puromycin (Life Technologies) (See supplementary materials for the antibiotic resistance genes used to integrate each genetic construct). Single clones were obtained using FACS sorting or limiting dilution. Single clones were chosen based on fluorescence or quantitative PCR for non-fluorescent constructs.

## Availability assay

The availability assay was based on the 'binding' assays in *Shimizu et al. (1999)* where soluble ligands bound to surface-expressed Notch2 receptors. Test cells were plated in 24-well plates (BD Falcon, San Jose, CA) at 25% confluence and treated with one of eight concentrations of 4-epiTc ranging from 0 to 200 ng/ml. In siRNA transfection experiments, silencing constructs were delivered after 24 hr of induction. After 48 hr of induction, cells from all 4-epiTc induction conditions were trypsinized, pooled into a single tube, and replated in triplicate at low (5–10% confluent) density. CHO-K1 and hN1(ΔICD)-Gal4esn cell lines were also plated as staining controls. After 4–6 hr, cells were blocked for 30 min at 37°C in blocking buffer (PBS with 2% FBS and 100 µg/ml $CaCl_2$). Next, cells were incubated with 10 µg/ml soluble Mouse Recombinant Dll1ext-Fc chimera (receptor availability) or Mouse Recombinant Notch1ext-Fc chimera (ligand availability), both from R&D Systems (5267-TK and 5026-DL, respectively, Minneapolis, MN) diluted in binding buffer (PBS with 2% Sigma bovine serum albumin and 100 µg/ml $CaCl_2$), for 1 hr at 4°C. After incubation, cells were washed three times with binding buffer and fixed with 4% methanol-free formaldehyde (Polysciences Inc., Warrington, PA). Cells were washed three times with binding buffer and permeabilized with 0.5% Triton X-100 (Thermo Scientific, Waltham, MA) and washed three more times.

Next, cells were blocked with blocking buffer for 30 min at room temperature and then incubated for 1 hr at room temperature with the following fluorescent secondary reagents: 1:500 dilution of anti-mouse IgG conjugated to Alexa 488 (Life Technologies) to stain cell-bound recombinant protein-Fc, 1:500 dilution of anti-GFP conjugated to Alexa 594 (Life Technologies) to visualize the ligand-CFP expressed by the cells, and a 1:10 dilution of HCS Cell Mask Blue (Life Technologies) to label the cells' cytoplasm for automatic segmentation. All reagents were diluted in binding buffer. Finally, cells were washed three times with binding buffer and mounted in 70% glycerol for microscopy analysis.

## Image acquisition and data analysis

Images were acquired with a CoolSnap HQ2 camera on a Nikon inverted TI-E microscope using a 20× long working distance objective. Metamorph 7.5 (Molecular Devices, Sunnyvale, CA) controlled the microscope, camera, stage (ASI instruments, Warren, MI) and brightfield and epifluorescence shutters (Sutter Instruments, Novato, CA) and collected the images. Fluorescent illumination was generated by the Sola LED light source (Lumencor, Beaverton, OR) and filtered through the Chroma filter sets SpGold, SpRed, and SpGreen. Brightfield illumination was generated by a halogen bulb.

Images were analyzed in Matlab 2012 (Mathworks, Natick, MA). First, cells were segmented on their labeling with the HCS Cell Mask Blue cytoplasmic stain. Next, total fluorescence in each fluorescence channel for each cell was calculated as follows. First, the value of the background fluorescence was computed in the neighborhood of the cell by taking the median of the unsegmented pixels in the neighborhood of the cell. Next, this background value was subtracted from each pixel's fluorescence value in the cell. Finally, all of the background-subtracted pixels were averaged to give the mean fluorescence for that cell. Image analysis code is posted on GitHub (https://github.com/llebon/image-analysis).

After this automatic processing, manual correction of the data was performed. This included imposing a gate on the segmented cell area to filter out multiple cells and segmentation errors. Next,

cells were screened by eye such that cells in physical contact with another cell were rejected, and only single, isolated cells were included in the analysis.

We found that for the same measured ligand-CFP level, we obtained different levels of surface availability, suggesting the ligands reach the cell surface with different efficiencies. To account for this difference, we normalized total ligand-CFP fluorescence in the Notch1+Dll1 and Notch1+Jag1 cell lines. Total ligand was plotted as 'Effective total ligand'.

We then plotted each cell's effective total ligand and availability fluorescence, and grouped cells into bins logarithmically-spaced along the effective total ligand axis. We plotted the median of these bins, and used a bootstrapped estimate of the median (MATLAB function bootci) to find the 95% confidence intervals of the bin median.

The snapshots in *Figure 2* were acquired on a Leica DMIRB/E fluorescence microscope with a 20× objective using the Chroma filter sets ECFP (31055v2) and EYFP (41028).

## Time-lapse imaging and analysis

Videos were performed as described previously in *Sprinzak et al. (2010)*. Cells were seeded onto glass-bottom plates (MatTek, Ashland, MA) coated with 5% fibronectin (Innovative Research, Novi, MI) in low-fluorescence imaging media, Alpha-MEM that includes 5% FBS and omits phenol red, riboflavin, folic acid, and vitamin B12 (Life Technologies, custom made). Cells were maintained at 37°C and 5% $CO_2$ in a chamber enclosing the microscope, an inverted Olympus IX81 equipped with Zero Drift Control (ZDC), a 20× NA 0.7 objective, and an iKon-M CCD camera (Andor, Belfast, NIR). All devices were controlled by Metamorph software.

Videos were analyzed in Matlab. Cell nuclei in each frame were identified automatically based on the CFP nuclear fluorescence, and the total fluorescence from each channel in each cell nuclei was recorded. Background subtraction was applied to each fluorescence value. In the plots, the average fluorescence from all of the cells in the frame is plotted vs time. Video analysis code is posted on GitHub (https://github.com/llebon/movie-analysis).

## Flow cytometry analysis

For analysis with flow cytometry, cells were dissociated with 0.25% trypsin (Life Technologies), diluted in FACS buffer (1× Hank's Balanced Salt Solution [Gibco] with 2.5 mg/ml BSA), and filtered through 40 µm strainers (BD Falcon). The cell suspension was screened for single-cell forward and side scatter and fluorescence intensity on a MacsQuant VYB instrument (Miltenyi Biotech, Bergisch Gladbach, Germany). Data was imported into Matlab 2012 for analysis. Analysis included imposing a gate on the forward and side-scatter area to omit dead cells and doublets and then analyzing the single-cell fluorescence intensity for each channel.

## Quantitative PCR

Quantitative PCR was used to confirm gene expression of non-fluorescent components. RNA was isolated from samples using the Qiagen RNAeasy kit according to the manufacturer's protocol. cDNA was synthesized from 1 µg of RNA using the iScript cDNA synthesis kit (Bio-Rad, Hercules, CA). For the real-time PCR reactions, 2 µl of cDNA was added to one reaction of SsoFast Probes Supermix (Bio-Rad). Each reaction was performed in triplicate. In parallel, three real-time PCR reactions were performed to measure β-actin levels in the sample, allowing us to compute a delta–delta CT value for the gene of interest in our cell lines. Reactions were performed on a Bio-Rad CFX Real-Time PCR Detection System.

Probe sets included the following:

### β-actin

> Primer 1: 5'-ACTGGGACGATATGGAGAAG-3',
> Primer 2: 5'-GGTCATCTTTTCACGGTTGG-3',
> Probe: 5'-HEX ACCACACCTTCTACAACGAGCTGC- Blk_FQ-3'

### Lfng

> Primer 1: 5'-GAAGTTCTGTCCCCTCGC-3'
> Primer 2: 5'-GATCCAGGTCTCGAACAGC-3'
> Probe: 5'-FAM ACTTTCTGGTGGTCTTGACGGCG-Blk_FQ-3'

## Mfng

> Primer 1: 5′-ACCACTCAAGTTTGTCCCAG-3′
> Primer 2: 5′-GATGAAGATGTCGCCTAGCTG-3′
> Probe: 5′-FAM TGAACCAACGGAACCCAGGACC-Blk_FQ-3′

## Rfng

> Primer 1: 5′-TCATTGCAGTCAAGACCACTC-3′
> Primer 2: 5′-CGGTGAAAATGAACGTCTGC-3′
> Probe: 5′-FAM CTCGTGAGATCCAGGTACGCAGC-Blk_FQ-3′

## siRNA delivery and sequences

siRNA against Notch1-ECD was delivered to cells using Lipofectamine 2000 reagent. Three dicer-substrate (dsiRNA from Integrated DNA Technologies, Coralville, IA) oligonucleotide duplexes against Notch1-ECD were pooled and 20 pmol of the mix were added to each well. IDT Universal Negative Control duplex was also transfected alongside each sample as a control.

### NECD Antisense 1

> 5′-rCrArG rCrGrA rGrCrA rCrUrC rArUrC rCrArC rGrUrC rCrUrG rGrCrU-3′
> 5′-rCrCrA rGrGrA rCrGrU rGrGrA rUrGrA rGrUrG rCrUrC rGrCT G-3′

### NECD Antisense 2

> 5′-rArCrA rCrCrA rGrUrG rCrArC rArArG rGrUrU rCrUrG rGrCrA rGrUrU-3′
> 5′-rCrUrG rCrCrA rGrArA rCrCrU rUrGrU rGrCrA rCrUrG rGrUG T-3′

### NECD Antisense 3

> 5′-rUrUrG rArUrC rUrCrG rCrArG rUrUrG rGrGrU rCrCrU rGrUrG rGrUrC-3′
> 5′-rCrCrA rCrArG rGrArC rCrCrA rArCrU rGrCrG rArGrA rUrCA A-3′

## Drosophila experiments

Animals were grown in standard food at room temperature (22°C) unless specified otherwise. The following fly strains were used in this study: (1) y w, (2) Dl⁹ᴾ/TM6, Tb¹, (3) N⁵⁵ᵉ¹¹/FM7c, (4) fng¹³/TM3, Sb¹, (5) fngᴸ⁷³/TM3, Sb¹, (6) Dp(3;3)bxd¹¹⁰/TM6, Tb¹, (7) Dlᴿᶠ/TM6C, Sb¹ Tb¹, (8) y¹ sc¹ v¹ P{nos-phiC31\int.NLS}X; P{CaryP}attP2, (Groth et al., 2004; Bischof et al., 2007), (9) y¹ M{vas-int.Dm}ZH-2A w*; PBac{y⁺-attP-3B}VK37 (Bloomington Drosophila Stock Center), (10) FRT 82B Serᴿˣ¹⁰⁶/TM6, Tb¹, (Venken et al., 2006), (11) Serʳᵉᵛ²⁻¹¹/TM3, Sb¹ Ser¹, Df(3R)Ser⁺ᴿ⁸²f²⁴ Tb¹/TM3, Sb¹ Ser¹, (Fleming et al., 1990), (12) P{Dlᵍᵗ⁻ʷᵗ}attP2, (13) PBac{Dlᵍᵗ⁻ʷᵗ}VK37, (14) PBac{Serᵍᵗ⁻ʷᵗ}VK37 (this study).

To generate a Delta genomic transgene, an 83.5-kb fragment containing the Delta locus and its flanking regions was transferred from BACR48K23 (BACPAC Resources Center, CHORI) to the attB-P[acman]-Apᴿ vector by using 'recombineering-mediated gap repair' (Venken et al., 2009). To generate a Delta genomic transgene, an 83.5-kb fragment containing the Delta locus and its flanking regions was transferred from BACR48K23 (BACPAC Resources Center, CHORI) to the attB-P[acman]-Apᴿ vector by using 'recombineering-mediated gap repair' (Muller et al., 2014). First, left and right homology arms (LA and RA, respectively) for the region of interest flanked by appropriate enzymes were generated by PCR using BACR48K23 as template and the following primers (restriction sites underlined):

| | |
|---|---|
| Dl-LA-Long-AscI-F | AGGCGCGCCATGCCATGACTGTTGTACTAACATAATGA |
| Dl-LA-Long-BamHI-R | AACGGATCCATATACCCTAGCTGTGCGGTAGTTCAT |
| Dl-RA-Long-BamHI-F | TTAGGATCCGCTGCGTGTGCTCTATAGGTGGTCATTA |
| Dl-RA-Long-PacI-R | CGGTTAATTAATCGGGATCGGCTTGCGGATCGTCAT |

The LA and RA were then cloned into the AscI-PacI digested attB- P[acman]-Ampᴿ vector via three-way ligation. The resulting construct was linearized by BamHI digestion and used to retrieve the 83.5-kb

fragment from *BACR48K23* by recombineering as described previously (*Groth et al., 2004*). All exons and exon-intron boundaries were verified by sequencing.

To generate a *Serrate* genomic transgene, a BAC clone with an 81.9-kb insert harboring the *Serrate* locus and its flanking regions was obtained from BACPAC Resources Center (*attB- P[acman]-CmR-CH321-69C08* [*Venken et al., 2009*]). ΦC31-mediated integration (*Venken et al., 2006*; *Bischof et al., 2007*)was used to insert the *Delta* and *Serrate* constructs into the *attP2* and *VK37* docking sites and to generate the *P{Dl^{gt-wt}}attP2* (*Dl^{gt-wt}*), *PBac{Dl^{gt-wt}}VK37* (*Dl^{gt-wt}*) and *PBac{Ser^{gt-wt}} VK37* (*Ser^{gt-wt}*) transgenes.

Adult wings were separated from anesthetized flies, incubated in 100% ethanol for 3 min, dried, and mounted in the DPX medium (Electron Microscopy Sciences, Hatfield PA). Wing images were obtained by using an AxioCam MRc5 camera mounted on a Zeiss Axioplan 2 microscope.

## Mathematical model of *cis* interactions

In order to interpret *cis* interaction measurements, we built on the model of interactions among receptors and ligands from *Sprinzak et al. (2010)*. This model was used to generate the schematic plots in *Figure 1A*, as well as to fit the single-cell data in *Figures 3 and 4*.

The model is based on two reactions between the Notch in cell *i* ($N_i$), and its interactions with ligands in the same cell ($D_i$), or a neighboring cell j ($D_j$),

$N_i + D_j \rightleftharpoons [N_iD_j] \rightarrow S_i$ *Trans*-activation, with association and dissociation rates $k_D^{\pm}$ and activation rate $k_S$.

$N_i + D_i \rightleftharpoons [N_iD_i] \rightarrow \varnothing$ *Cis*-inhibition, with association and dissociation rates $k_C^{\pm}$ and inactivation rate $k_I$.

Notch is created with a constant rate $\beta_N$ and degraded at a linear rate $\gamma_N N_i$. Ligand is produced at a constant rate $\beta_D$ and degraded with a linear rate $\gamma_D D_i$. $N_iD_j$ represents the complex of a Notch receptor in cell *i* bound to a ligand in cell *j*. When *i = j* this terms describes a *cis* interaction. Because we chose cell-plating conditions to isolate *cis* interaction, we can ignore the *trans*-activation terms.

These two reactions can be rewritten as a set of ordinary differential equations:

$$\frac{d[N_i]}{dt} = \beta_N - \gamma_N N_i - (k_C^+ N_i D_i - k_c^- [N_iD_i])$$

$$\frac{d[D_i]}{dt} = \beta_D - \gamma_D D_i - (k_C^+ N_i D_i - k_c^- [N_iD_i])$$

$$\frac{d[N_iD_i]}{dt} = k_C^+ N_i D_i - k_C^- [N_iD_i] - k_I [N_iD_i].$$

We assume that the bound *cis*-complex achieves a quasi-steady state $\left(\frac{d[N_iD_i]}{dt} \approx 0\right)$. Using this assumption, we derive the following relationship:

$$N_{steady\,state} = \frac{\dfrac{\beta_N}{\gamma_N}}{1 + \dfrac{D_{steady\,state}}{k_c \gamma_N}}$$

where $k_c^{-1}$ is defined as $\dfrac{k_C^+ k_I}{k_C^- + k_I}$. The physical meaning of this expression is that available Notch receptor is a decreasing function of *cis*-Delta concentration. When there is no *cis*-Delta expression, steady state receptor levels are $\beta_N / \gamma_N$. However, as *cis*-Delta increases, the level of available Notch drops below this maximal value. The amount of Delta necessary to deplete Notch receptor to one half of its maximal concentration in the absence of *cis*-Delta is $k_c \gamma_N$.

## Acknowledgements

This work was supported by the Gordon and Betty Moore Foundation through Grant GBMF2809 to the Caltech Programmable Molecular Technology Initiative; the National Science Foundation under

Grant No. EFRI 1137269 and The NIH under grant R01 HD705335. We also acknowledge support from the NIH/NIGMS (R01GM084135 to HJN) and the March of Dimes Foundation (#1-FY10-501 to HJN). We thank Pulin Li, Joseph Markson, Sandy Nandagopal, Amit Lakhanpal, Emily Capra, Fangyuan Ding, and Leah Santat for technical assistance, as well as discussions and comments, Gerry Weinmaster for helpful comments on the manuscript, Yi-Dong Li and Jessica Leonardi for assistance with the generation of transgenic flies, and Robert Fleming, Shinya Yamamoto, and The Bloomington *Drosophila* Stock Center for fly strains.

## Additional information

### Funding

| Funder | Author |
| --- | --- |
| Gordon and Betty Moore Foundation | Lauren LeBon, Michael B Elowitz |
| National Science Foundation | Lauren LeBon, Michael B Elowitz |
| National Institutes of Health | Tom V Lee, Hamed Jafar-Nejad |
| March of Dimes Foundation | Tom V Lee, Hamed Jafar-Nejad |

The funders had no role in study design, data collection and interpretation, or the decision to submit the work for publication.

### Author contributions

LL, TVL, DS, Conception and design, Acquisition of data, Analysis and interpretation of data, Drafting or revising the article; HJ-N, MBE, Conception and design, Analysis and interpretation of data, Drafting or revising the article

## Additional files

### Supplementary file

• Supplementary file 1. (**A**) Cell lines used in this work. (**B**) Plasmids used in this work.

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
