## [Decision Letter]

Thank you for sending your work entitled “Fringe proteins modulate Notch-ligand cis and trans interactions to specify signaling states” for consideration at *eLife.* Your article has been favorably evaluated by Fiona Watt (Senior editor) and 3 reviewers, one of whom is a member of our Board of Reviewing Editors.

The Reviewing editor and the other reviewers discussed their comments before we reached this decision, and the Reviewing editor has assembled the following comments to help you prepare a revised submission.

All the reviewers felt that the tissue culture analysis section of the study was elegant and carefully conducted, and showed a novel role for Fringe proteins in regulation of cis interactions. The reviewers felt, however, that the studies in the wing were less compelling, and many of the conclusions needed to be qualified for this section. There were also some concerns about the effect of possibly masking PDZ domain interactions. In detail, the reviewers felt that:

1) Some of the effects in *Drosophila* are very weak. For example, the authors attempt to identify phenotypes indicative of effects of fng on cis interactions. For Ser, the key comparison is the genotype shown in 6I vs 6E and G. It looks like a simple additive effect. Also, the quantitation of the E vs I genotypes (5th vs 9th columns in K), shows that for 95% of the flies there is no effect, and for 5% there is a shift of one “class” I would guess that this is within the range of what one could see just by changes in genetic background, more rigorous experiments would be needed to make a convincing statement as to whether one is seeing effects of Fng on cis interactions here.

2) To interpret the effects of changing gene dosage as affecting cis versus trans effects, one also has to be mindful of the expression patterns of all of these genes relative to where Notch activation is occurring. Each of the players they are examining (Delta, Serrate, Fringe), have complex and dynamic expression patterns with respect to both the wing margin and the veins, but this aspect is ignored. This caveat should be discussed in the revised submission.

3) The in vitro assay is very elegant; however, it seems like most (if not all) experiments where performed using ligand fusions. The nature of these fusions should be discussed a little more in the text. As ligands bind to PDZ proteins (which often require binding motifs to be presented at or near the C-terminus, one might imagine that C-terminal fusion proteins could alter ligand trafficking). If this point does not apply, or if this reviewer has misunderstood the nature of the fusions involved, then textual changes to make this clear should be considered. If this issue is not controlled for, then we would recommend the authors should repeat one or two of the assays with completely wild type ligands, just to show that their conclusions are still valid in this context.

Minor comments:

1) A weakness in the authors’ interpretation of their results comes out where they make statements like “This suggests the intriguing possibility, which will need to be tested in additional contexts, that the Notch pathway architecture permits heterotypic signaling (between cells in different states) but not homotypic signaling (between cells in the same state).” And “cis-inhibition of Delta could thus explain why dorsal cells do not signal to one another, even though they express and are responsive to Delta.”

In fact, there is evidence in the published literature, based on analysis of Delta mutant clones, that dorsal cells do signal to each other through Delta (in addition to receiving a Delta signal from ventral cells). See [19]; [12]; and [30].

2) “Troost and Klein recently examined signaling at the boundary with higher time resolution than in previous work. They found that signaling occurred in two sequential phases (15): First, dorsal cells signal via Serrate to ventral cells. Subsequently, ventral cells up-regulate Delta and signal back to dorsal cells.”

Actually, that's been known for at least 15 years before the Troost & Klein paper, e.g., see [33].

---

## [Author Response]

We thank the editors and reviewers for their thoughtful comments on the manuscript. We have performed additional experiments and made other changes to the manuscript and figures to address their concerns.

In particular:

We added a new figure (Figure 3—figure supplement 4) to address the effects of fluorescent protein fusions to DSL ligands.

We constructed and analyzed additional *Drosophila* mutants to address the reviewer comments, as shown in the revised Figure 6.

We updated the cartoons in Figures 7 and 8 and their associated captions for clarity.

We have also made smaller changes to text to improve clarity and readability.

*1) Some of the effects in* Drosophila *are very weak. For example, the authors attempt to identify phenotypes indicative of effects of fng on cis interactions. For Ser, the key comparison is the genotype shown in 6I vs 6E and G. It looks like a simple additive effect. Also, the quantitation of the E vs I genotypes (5th vs 9th columns in K), shows that for 95% of the flies there is no effect, and for 5% there is a shift of one “class” I would guess that this is within the range of what one could see just by changes in genetic background, more rigorous experiments would be needed to make a convincing statement as to whether one is seeing effects of Fng on cis interactions here*.

The results we present in Figure 6 are surprising when one considers the alternative hypothesis: if it were the case that Fringe only affected Serrate-Notch trans activation, then removing a copy of fringe in the 1X Notch 3X Serrate background would be expected to increase Serrate-Notch signaling and rescue the wing margin loss phenotype observed in 1X Notch 3X Serrate animals, and perhaps even result in wing outgrowth. Instead, what we observed was a worsening of the wing margin loss phenotype, consistent with a further loss of Notch signaling. The phenotypes observed in the 1X Notch 3X Serrate 1X fringe animals are not simply an addition of the 1X Notch 1X fringe and 1X Notch 3X Serrate phenotypes; instead, we observe a qualitatively new phenotype – the “Class 3” L3 and L4 wing margin loss, that we had previously only seen among 1X Notch 4X Serrate animals. This defect is never observed in the 1X Notch 3X Serrate animals or in 1X Notch 1X fringe animals. Thus, the effect of Fringe loss in this background is only consistent with a picture where Fringe blocks the cis inhibition of Notch signaling by Serrate.

To show that these effects are not specific to the genetic background of the animals tested in these experiments, we performed additional studies on the effects of Fringe. We examined the outcomes of removing a copy of fringe from 2X (wild-type) Notch 4X Serrate or 2X Notch 4X Delta flies. With two copies of fringe, these animals exhibit no wing margin abnormalities. However, removing a copy of fringe from the 4X Serrate animals results in Notch loss of activity phenotypes at the wing margin which was more severe than 3X Serrate 1X fringe animals shown in the original manuscript (Figure 6). By contrast, removing a copy of fringe from the 4X Delta animals did not result in wing margin phenotypes. To reiterate the point above: if Fringe only affected Serrate-Notch trans activation, one would expect that removing a copy of fringe would lead to gain of Notch signaling defects, such as wing outgrowth, due to an increase in Serrate-Notch signaling. But not only did we not observe any gain of Notch signaling effects, we instead observed loss of Notch signaling phenotypes in a proportion of the animals that worsens as we increase the Serrate gene dosage. These results are consistent with a model in which the normal role of Fringe is to block Serrate-Notch cis-inhibition.

The new data are shown in an expanded Figure 6. We have also added additional text to the manuscript to better explain these data and the points above.

*2) To interpret the effects of changing gene dosage as affecting cis versus trans effects, one also has to be mindful of the expression patterns of all of these genes relative to where Notch activation is occurring. Each of the players they are examining (Delta, Serrate, Fringe), have complex and dynamic expression patterns with respect to both the wing margin and the veins, but this aspect is ignored. This caveat should be discussed in the revised submission*.

We agree and we have added text in the revised manuscript to describe the dynamics of expression of each of the components in both wing margin and wing veins and to discuss this caveat. We emphasize that wing vein formation mainly relies on expression of the Delta ligand, while wing margin formation relies on expression of both DSL ligands as well as feedback from Notch target genes.

*3) The in vitro assay is very elegant; however, it seems like most (if not all) experiments where performed using ligand fusions. The nature of these fusions should be discussed a little more in the text. As ligands bind to PDZ proteins (which often require binding motifs to be presented at or near the C-terminus, one might imagine that C-terminal fusion proteins could alter ligand trafficking). If this point does not apply, or if this reviewer has misunderstood the nature of the fusions involved, then textual changes to make this clear should be considered. If this issue is not controlled for, then we would recommend the authors should repeat one or two of the assays with completely wild type ligands, just to show that their conclusions are still valid in this context*.

To make sure this potential issue is clear in the manuscript, we now explicitly describe potential problems with using ligand-fusion proteins in the Results section.

More importantly, we added a new figure and experiments (Figure 3—figure supplement 4) to address this issue. We directly compared the effects of transient expression of tagged and untagged ligands on Notch1 availability. We found that both the CFP-tagged ligands and untagged ligands fully reduced Notch1 availability to background levels, indicating that both ligands are capable of cis inhibition. Further, co-expression of Lfng preserved the ability of both tagged and untagged Dll1 to cis-inhibit Notch1, but weakened both tagged and untagged Jag1 cis-inhibition of Notch1. These results indicate that these key qualitative conclusions of the paper hold for untagged as well as tagged ligands. Together, these results show that the tagged ligand-fluorescent protein fusions proteins behave qualitatively similarly to their untagged counterparts, at least for these experiments.

We have also added some clarification in the methods section to describe the construction of the ligand-FP fusions (Table 2).

Minor comments:

*1) A weakness in the authors interpretation of their results comes out where they make statements like “This suggests the intriguing possibility, which will need to be tested in additional contexts, that the Notch pathway architecture permits heterotypic signaling (between cells in different states) but not homotypic signaling (between cells in the same state).” And “cis-inhibition of Delta could thus explain why dorsal cells do not signal to one another, even though they express and are responsive to Delta*.*”*

*In fact, there is evidence in the published literature, based on analysis of Delta mutant clones, that dorsal cells do signal to each other through Delta (in addition to receiving a Delta signal from ventral cells). See*
[19]*;*
[12]*; and*
[30]*.*

Thank you for raising this issue. In the Doherty et al. reference (an important paper that we now cite in the manuscript), the authors reported that ectopic Dl expression could activate Notch in dorsal cells. However, loss of endogenous Dl in dorsal compartment clones did not lead to any wing margin defects, suggesting that dorsal Dl signaling does not play a critical role in wing development. [12], presented robust evidence for activation of Notch target genes by ectopic Delta-expressing clones on the dorsal side of the boundary, but no indication for a signaling activity of endogenous Dl in the dorsal compartment. Both of these results are consistent with the picture presented in our Figures 7 and 8.

In contrast, the Lei et al paper includes one mutant clone experiment suggesting that endogenous Delta expression in dorsal cells of the wing disc contributes to increases in Serrate expression in dorsal cells, i.e. that endogenous, presumably homotypic, Delta-Notch signaling occurs among the dorsal cells and contributes to Serrate upregulation in the dorsal compartment. In their Figure 4 we see 3 Dl- clones. Clone 1 is large, spans both the dorsal and ventral compartments, and loses Serrate expression, most likely because the positive feedback loop between ventral Delta and dorsal Serrate is lost. Clone 2 is small and does not affect Serrate expression. Clone 3 is dorsal and seems to mildly affect Serrate expression. The authors interpret this as Delta trans-activation in the dorsal compartment. However, the effect is weak at best for a few reasons: First, the loss of Delta expression on the dorsal side does not lead to any wing margin defects in their work, indicating no physiological role for this putative homotypic signaling. Second, you can still see Serrate expression in the clone, indicating that it is at most a partial effect. Third, it appears to be based on only a single clone. Thus, the results in Lei et al do not provide strong evidence for signaling among dorsal cells and suggest that if it does occur it is certainly a less important effect than the heterotypic signaling between dorsal and ventral during boundary formation.

In the revised manuscript we explicitly discuss these issues and cite these references to acknowledge this work, while also emphasizing that heterotypic signaling plays a much more important and non-redundant role in wing margin formation (see text and caption to Figure 8). Most importantly, all evidence still indicates that the effects of Fringe on cis interactions between Notch and ligands is a key mechanism for the dominance of heterotypic signaling over homotypic signaling in this developmental context.

*2) “Troost and Klein recently examined signaling at the boundary with higher time resolution than in previous work. They found that signaling occurred in two sequential phases: First, dorsal cells signal via Serrate to ventral cells. Subsequently, ventral cells up-regulate Delta and signal back to dorsal cells*.*”*

*Actually, that's been known for at least 15 years before the Troost & Klein paper, e.g., see*
[33].

The Micchelli et al. paper does describe sequential signaling at the boundary through the dynamic expression of Cut and Wingless, and the effects of these target genes on Notch activity. Cut expression is activated by Notch signaling at the dorsal ventral wing boundary, and is maintained indirectly by Wingless via upregulation of Delta and Serrate in cells flanking the boundary.

However, our model focuses on the earliest step in this process, prior to the emergence of Cut expression; the first Notch signal exchanged between dorsal and ventral cells. In the Micchelli et al. reference this initial step is described as “reciprocal.” The Doherty et al. paper suggests and the Troost & Klein paper demonstrates that this initial reciprocal step is in fact two sequential steps. Our original manuscript omitted describing these signaling events in the later stages, which might have led to confusion between the events in the early and late stages of boundary formation. In the new manuscript, we have added additional text describing the late stages of boundary formation and have tried to clearly specify which stage we are talking about.